# In vitro analysis of RQC activities provides insights into the mechanism and function of CAT tailing

Beatriz A Osuna[1,2], Conor J Howard[2,3,4], Subheksha KC[1], Adam Frost[2,3,4,5]*, David E Weinberg[1,6]*

[1]Department of Cellular and Molecular Pharmacology, University of California, San Francisco, San Francisco, United States; [2]Department of Biochemistry and Biophysics, University of California, San Francisco, San Francisco, United States; [3]California Institute for Quantitative Biomedical Research, San Francisco, United States; [4]Chan Zuckerberg Biohub, San Francisco, United States; [5]Department of Biochemistry, University of Utah, Salt Lake City, United States; [6]Sandler Faculty Fellows Program, University of California, San Francisco, San Francisco, United States

**Abstract** Ribosomes can stall during translation due to defects in the mRNA template or translation machinery, leading to the production of incomplete proteins. The Ribosome-associated Quality control Complex (RQC) engages stalled ribosomes and targets nascent polypeptides for proteasomal degradation. However, how each RQC component contributes to this process remains unclear. Here we demonstrate that key RQC activities—Ltn1p-dependent ubiquitination and Rqc2p-mediated Carboxy-terminal Alanine and Threonine (CAT) tail elongation—can be recapitulated in vitro with a yeast cell-free system. Using this approach, we determined that CAT tailing is mechanistically distinct from canonical translation, that Ltn1p-mediated ubiquitination depends on the poorly characterized RQC component Rqc1p, and that the process of CAT tailing enables robust ubiquitination of the nascent polypeptide. These findings establish a novel system to study the RQC and provide a framework for understanding how RQC factors coordinate their activities to facilitate clearance of incompletely synthesized proteins.

*For correspondence: adam. frost@ucsf.edu (AF); david. weinberg@ucsf.edu (DEW)

**Competing interests:** The authors declare that no competing interests exist.

## Introduction

Eukaryotic cells contain several cotranslational quality-control pathways that limit the production of aberrant proteins and thereby maintain protein homeostasis. One such pathway is activated when a ribosome fails to complete translation, leading to the recruitment of specialized factors that disassemble the stalled ribosome and facilitate degradation of the nascent protein (*Brandman and Hegde, 2016*; *Shoemaker and Green, 2012*). A key effector of this process is the highly conserved Ribosome-associated Quality control Complex (RQC), which in budding yeast comprises the E3 ubiquitin ligase Ltn1p, the ATPase Cdc48p, and the poorly characterized proteins Rqc1p and Rqc2p (*Brandman et al., 2012*; *Defenouillère et al., 2013*; *Verma et al., 2013*)—the human homologs of which are Listerin, VCP/p97, TCF25, and NEMF, respectively. The stalled translation complex is first separated into subunits by ribosome splitting factors, allowing the small ribosomal subunit (40*S*) and mRNA to be released. The RQC then recognizes and assembles on the large ribosomal subunit (60*S*) that still contains a nascent polypeptide linked to a tRNA molecule (60*S*:peptidyl–tRNA). Ltn1p facilitates ubiquitination of the nascent chain while on the 60*S* subunit, marking the incompletely synthesized protein for proteasomal degradation (*Bengtson and Joazeiro, 2010*; *Shao et al., 2013*).

**eLife digest** Cells make proteins by reading instructions encoded in molecules called messenger RNAs. Structures called ribosomes move along the messenger RNAs and translate the coded instructions to build new proteins from building blocks known as amino acids. Normally, a ribosome will encounter a stop signal on the messenger RNA, which ends translation and allows the newly built protein to be released. Sometimes, however, ribosomes stall before they reach the genuine stop signal, which can happen due to defects in the messenger RNAs or ribosomes.

To prevent incomplete proteins from accumulating and causing damage, cells contain a group of other proteins called the Ribosome-associated Quality-control Complex (or RQC for short). This quality-control complex is composed of three components that assemble on stalled ribosomes and attach two different tags to the incomplete protein. One component adds a degradation tag called ubiquitin. A second component works with the ribosome to tag the incomplete protein with a 'tail' that contains the amino acids alanine and threonine. These amino acids are abbreviated to A and T, and are added to the end of the protein known as the 'C-terminus', so this tag is named a 'CAT tail'. Although all three RQC components are needed to degrade incomplete proteins, little was known about why the CAT tails are added, or what the third component – a protein called Rqc1p – actually does.

Osuna et al. have now investigated how the apparently unrelated activities of RQC components are coordinated to destroy incomplete proteins. Ribosomes and RQC components from yeast cells were extracted and mixed in the laboratory with a messenger RNA that stalls ribosomes. In this cell-free system, the RQC components could still tag incomplete proteins with both ubiquitin and CAT tails. Osuna et al. then used this system to show that the way the ribosome added amino acids to form a CAT tail was different from how it normally builds proteins. The experiments also showed that in order for the ubiquitin tags to be added efficiently, Rqc1p must be present, and in some cases, the incomplete proteins also need to be 'CAT tailed'.

When either were missing, very few ubiquitin tags could be added to the incomplete proteins. The results show that the three core RQC components need to work together to degrade incomplete proteins. This quality-control complex is also found in mice and humans, and mice with mutations in the genes that encode RQC components often have damaged nervous systems. In the future, researchers building upon these findings and other studies of the RQC may eventually understand the relationship between the RQC and neurodegenerative diseases in humans.

Additionally, Rqc2p recruits charged tRNAs to the 60$S$ subunit to direct elongation of the nascent protein with a Carboxy-terminal Alanine and Threonine extension, or CAT tail (*Shen et al., 2015*).

Structural analysis of the yeast RQC, identification of the tRNA molecules that co-purify with the RQC, and biochemical characterization of failed nascent chains suggested that CAT tailing occurs on the 60$S$ subunit by a unique mechanism that does not require an mRNA template or the 40$S$ subunit (*Shen et al., 2015*). However, many questions about the mechanism of CAT-tail synthesis and the consequences of elongating nascent polypeptides with CAT tails remain unanswered. Recent studies have suggested that one function of CAT tails is to facilitate aggregation of nascent polypeptides that fail to be ubiquitinated by Ltn1p (due to either disruptions in *LTN1* or the absence of a suitable ubiquitin acceptor). CAT tail-driven aggregation may limit the otherwise toxic effects of incomplete translation products accumulating in the cytoplasm (*Choe et al., 2016*; *Defenouillère et al., 2016*; *Yonashiro et al., 2016*). However, our understanding of the functions of CAT tails in the context of an intact RQC or of the process of CAT tailing itself remains incomplete.

Previous studies have analyzed the RQC in vitro by using cell-free translation systems based on rabbit reticulocyte lysates (*Shao et al., 2013*) or *Neurospora crassa* extracts (*Doamekpor et al., 2016*). In the presence of a suitable mRNA substrate, both cell-free systems recapitulate Ltn1p-dependent ubiquitination and thereby provide valuable insight into the mechanism by which Ltn1p orthologs discriminate between elongating and stalled ribosomes (*Shao et al., 2013*) and the role of the N-terminal domain of Ltn1p in binding the 60$S$ subunit (*Doamekpor et al., 2016*). However, neither system recapitulates Rcq2p-dependent CAT tailing, leaving important unanswered questions

about how CAT tails are synthesized and whether the two principal activities of the RQC—ubiquitination by Ltn1p and CAT tailing by Rqc2p—are functionally related.

Although many studies have identified Rqc1p/TCF25 as a core component of the yeast and mammalian RQC required for nascent-chain degradation (*Brandman et al., 2012*; *Defenouillère et al., 2013*; *Shao and Hegde, 2014*), Rqc1p's precise structural and functional roles in the complex remain unclear. Previous work in yeast suggested that Rqc1p acts after Ltn1p to promote nascent-chain degradation. This hypothesis emerged from two lines of evidence: The presence of polyubiquitinated proteins in purified RQC depends on Ltn1p (and to a lesser extent on Rqc2p) but not on Rqc1p or Cdc48p (*Brandman and Hegde, 2016*; *Brandman et al., 2012*); and recruitment of Cdc48p to the 60S subunit requires Rqc1p and nascent-chain ubiquitination (*Defenouillère et al., 2013*). However, these studies did not determine whether Rqc1p is necessary for ubiquitination of the nascent chain itself or whether recruitment of Cdc48p requires a direct interaction with Rqc1p. Therefore, the mechanism by which Rqc1p promotes nascent-chain degradation in vivo has remained unclear.

In this study, we provide an in vitro characterization of the RQC in a budding-yeast extract that uniquely recapitulates ubiquitination by Ltn1p and CAT tailing by Rqc2p, providing new insights into RQC action in promoting degradation of stalled translation products.

## Results

### A cell-free system that recapitulates Rqc2p-mediated nascent-chain elongation

Because CAT tails have thus far only been observed in *S. cerevisiae* (*Choe et al., 2016*; *Defenouillère et al., 2016*; *Shen et al., 2015*; *Yonashiro et al., 2016*), we used *S. cerevisiae* extracts to recapitulate Rqc2p-mediated elongation in vitro. Although *S. cerevisiae* has long been used for in vitro translation (*Hussain and Leibowitz, 1986*; *Iizuka et al., 1994*; *Rojas-Duran and Gilbert, 2012*; *Tarun and Sachs, 1995*), these reactions are notoriously inefficient. Further exacerbating this problem, we aimed to program these reactions with truncated mRNA substrates that trigger quality control, which are translated less efficiently because they lack poly(A) tails that normally enhance translation. Thus, we found it necessary to first establish an optimized protocol that could reproducibly generate translation products that were detectable by immunoblotting (see Materials and methods). Critical aspects of our protocol included: (1) lysing cells with a freezer mill under cryogenic conditions rather than by bead beating in the cold; (2) minimizing the number of lysis cycles; (3) removing small molecules by dialysis rather than by size-exclusion chromatography; and (4) programming translation reactions with an mRNA encoding a small protein (i.e., 23 kDa NanoLuc luciferase), which is translated more efficiently than an mRNA encoding a larger protein (e.g., 62 kDa firefly luciferase).

To produce a substrate for the RQC, we used a truncated reporter mRNA that terminates with a sense codon (i.e., does not contain a stop codon, 3′–untranslated region (3′-UTR), or poly(A) tail). This type of mRNA substrate has been shown previously to generate a ribosome–nascent chain complex stalled at the 3′ end of the message (*Becker et al., 2011*; *Shao et al., 2013*). Our reporter mRNA encodes a NanoLuc luciferase (NL) protein in which the seven native lysine residues have been mutated to arginine to avoid potential confounding effects of lysine ubiquitination. The protein also includes an N-terminal 3xHA tag (which is naturally devoid of lysines) to allow detection of translation products by immunoblotting.

Programming *S. cerevisiae* in vitro translation (ScIVT) reactions with a full-length control mRNA that contains a stop codon and 3′-UTR resulted in the time-dependent accumulation of a 23 kDa product corresponding to 3xHA-NL (*Figure 1A*, left). In contrast, ScIVT of a truncated mRNA initially produced a ~43 kDa mass-shifted product not observed in control reactions (*Figure 1A*, right), which we hypothesized corresponded to a peptidyl–tRNA intermediate. Remarkably, as the reaction proceeded we observed the disappearance of the initial ~43 kDa product and concomitant accumulation of smaller mass-shifted products ranging from 23 kDa to 43 kDa (*Figure 1A*, right). These mass-shifted products were reminiscent of CAT-tailed species previously observed in vivo (*Choe et al., 2016*; *Defenouillère et al., 2016*; *Shen et al., 2015*; *Yonashiro et al., 2016*).

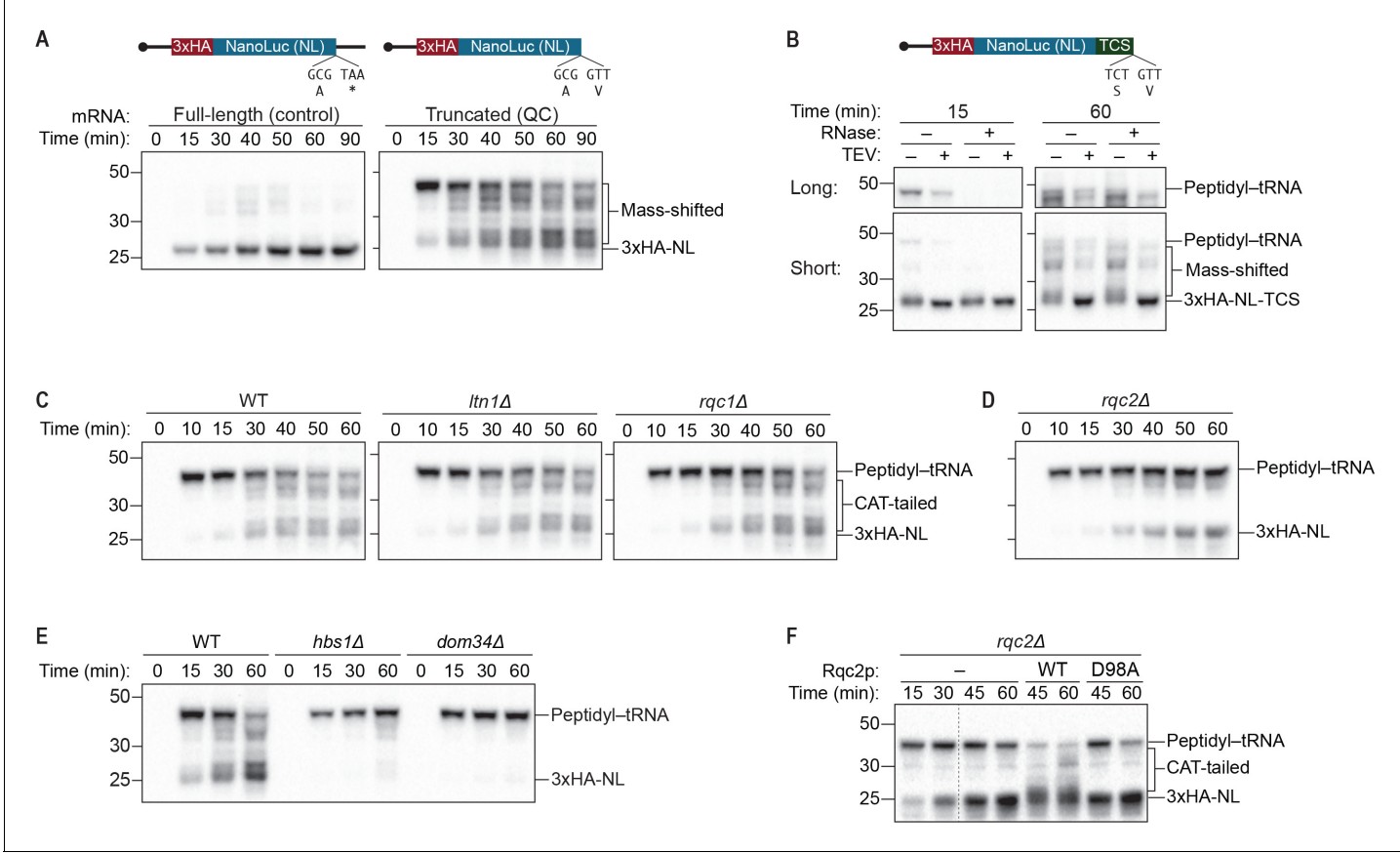

**Figure 1.** An *S. cerevisiae in vitro* translation system recapitulates synthesis of Rqc2p-dependent polypeptide extensions. (**A**) Time courses of *S. cerevisiae in vitro* translation (ScIVT) reactions. ScIVT reactions were prepared using wild-type (WT) extracts and 1 µg of either a full-length (left; includes a stop codon and 3'-UTR) or truncated (right; encodes a terminal valine residue) mRNA encoding lysine-free 3xHA-NanoLuc (3xHA-NL). At the indicated time points, aliquots of the reactions were quenched in 2X Laemmli Sample Buffer. Proteins were separated by SDS-PAGE, and HA-tagged translation products were visualized by immunoblotting. (**B**) Analyses of mass-shifted products. An ScIVT reaction was prepared using WT extracts and a lysine-free truncated mRNA substrate that also encodes a TEV cleavage site (TCS). Translation was halted after 15 or 60 min by addition of 20 mM EDTA, after which reactions were treated without (–) or with (+) TEV and/or RNase A/T1 cocktail for 60 min. Translation products were analyzed by immunoblotting as in (**A**). 'Long' and 'Short' refer to exposure times of the blots. (**C–E**) Genetic analysis of mass-shifted products. ScIVT reactions were prepared using extracts from strains of the indicated genotypes and a lysine-free truncated mRNA substrate. Reactions were performed and analyzed as in (**A**) but with less mRNA (480 ng). The species that migrate just below the peptidyl–tRNA in *rqc2Δ* extracts in (**D**) represent peptidyl–tRNA degradation products that arise due to prolonged incubation in the absence of Rqc2p. (**F**) Rescuing Rqc2p deficiency in vitro. ScIVT reactions were prepared using *rqc2Δ* extracts and a lysine-free truncated mRNA substrate. After 30 min of translation, reactions were supplemented with either protein storage buffer (–) or purified Rqc2p (WT or CAT-tailing deficient D98A at 420 nM final concentration) and indicated time points were analyzed by SDS-PAGE and immunoblotting. Dashed lines indicate where intervening lanes were removed for clarity.

The following figure supplement is available for figure 1:

**Figure supplement 1.** Purified wild-type and mutant Rqc2p.

To further characterize the mass-shifted species, we added a sequence encoding a TEV protease cleavage site (TCS) at the 3' end of our mRNA substrate (*Shen et al., 2015*) and determined the susceptibility of the mass-shifted species to TEV protease and RNase. We reasoned that a nascent polypeptide with its C-terminus covalently linked to a tRNA molecule would be liberated by either TEV protease or RNase. In contrast, a nascent polypeptide containing an untemplated C-terminal amino-acid extension (e.g., a CAT tail) would be cleaved by TEV protease but not RNase, and a protein containing additional mass due to modifications anywhere except the C-terminus would be unaffected by either treatment. When treated with either RNase or TEV protease, the ~45 kDa intermediate observed at early time points was converted to a 25 kDa species, corresponding to the

molecular weight of 3xHA-NL-TCS (*Figure 1B*). Given that the average molecular weight of a tRNA is ~20 kDa, these results suggest that the ~45 kDa species contained a tRNA covalently linked to the C-terminus of 3xHA-NL-TCS ('peptidyl–tRNA')—which has previously been characterized as an intermediate of the quality-control pathway (*Shao et al., 2013*; *Shoemaker et al., 2010*; *Tsuboi et al., 2012*). In contrast, the heterogeneous collection of ~25–45 kDa products that accumulated at later time points were only affected by TEV protease, converting them to a discrete 25 kDa species (*Figure 1B*), as expected if the products originally contained additional mass downstream of the TCS (i.e., appended to the C-terminus of 3xHA-NL-TCS). Because the truncated mRNA substrate used for ScIVT contained no sequences downstream of the TCS, this additional mass was necessarily untemplated and therefore consistent with CAT tails. Notably, the prominent peptidyl–tRNA species that accumulated at early time points was largely absent after 60 min of translation (*Figure 1A and B*), presumably due to peptidyl–tRNA hydrolysis that occurred after untemplated elongation of the nascent chain.

In addition to being C-terminal and untemplated, another known feature of CAT tails is that their synthesis is strictly dependent on Rqc2p but not on Ltn1p or Rqc1p (*Choe et al., 2016*; *Defenouillère et al., 2016*; *Shen et al., 2015*; *Yonashiro et al., 2016*). To determine if the ~23–43 kDa mass-shifted species share this property, we took advantage of the genetic tractability of *S. cerevisiae* and non-essential nature of the RQC by performing ScIVT using extracts prepared from *ltn1Δ*, *rqc1Δ*, and *rqc2Δ* strains. While reactions using *ltn1Δ* and *rqc1Δ* extracts yielded all of the mass-shifted products observed when using wild-type (WT) extracts (*Figure 1C*), reactions using *rqc2Δ* extracts did not (*Figure 1D*), consistent with those species corresponding to CAT-tailed protein. However, rather than producing the expected 23 kDa 3xHA-NL protein, reactions lacking Rqc2p generated a relatively stable 43 kDa peptidyl–tRNA species, indicating a defect in peptidyl–tRNA hydrolysis. This finding suggests that in addition to facilitating the incorporation of untemplated amino acids, Rqc2p may also be involved in promoting hydrolysis of the final peptidyl–tRNA bond and thereby liberating the nascent polypeptide.

Previous biochemical and structural studies have suggested that Rqc2p engagement and subsequent CAT tailing must be preceded by ribosome splitting, which exposes the P-site tRNA and the surface of the Sarcin-Ricin loop (SRL) that are recognized by Rqc2p/NEMF (*Lyumkis et al., 2014*; *Shao et al., 2015*; *Shen et al., 2015*). In the case of truncated mRNAs that generate a ribosome stalled at the mRNA 3′ end with an empty A site, ribosome splitting is effected by the release-factor mimics Hbs1p and Dom34p (*Shao et al., 2013*; *Shoemaker et al., 2010*). Accordingly, ScIVT of a truncated mRNA in *hbs1Δ* or *dom34Δ* extracts generated only the ~43 kDa product corresponding to an especially stable peptidyl–tRNA (*Figure 1E*), which is presumably protected within a stalled but intact 80*S* ribosome. Collectively, these data demonstrate that ScIVT of a truncated mRNA generates polypeptides containing untemplated Rqc2p-dependent C-terminal extensions. Although we have not been able to confirm that these extensions are composed of alanine and threonine residues (for technical reasons), we suspect that this is the case and therefore refer to the extensions as CAT tails for simplicity.

The lack of CAT-tailing activity in *rqc2Δ* extracts (*Figure 1D*) is consistent with previous observations that CAT tails are absent from *rqc2Δ* strains in vivo (*Choe et al., 2016*; *Defenouillère et al., 2016*; *Shen et al., 2015*; *Yonashiro et al., 2016*), which could reflect either a direct role for Rqc2p in CAT tailing (as suggested by structural studies) or indirect effects of *RQC2* disruption on CAT tailing. To distinguish between these possibilities, we tested whether the absence of CAT-tailing activity in *rqc2Δ* extracts could be rescued by adding purified Rqc2p to ScIVT reactions already in progress. Remarkably, the addition of exogenous Rqc2p (*Figure 1—figure supplement 1*) to *rqc2Δ* extracts restored both CAT-tail synthesis and peptidyl–tRNA hydrolysis, whereas the addition of a CAT-tailing-deficient Rqc2p mutant containing the D98A substitution (*Shen et al., 2015*; *Yonashiro et al., 2016*) did not promote either CAT-tail synthesis or robust peptidyl–tRNA hydrolysis (*Figure 1F*). These results provide direct evidence that Rqc2p is biochemically required for CAT tailing, consistent with its proposed role in recruiting alanine- and threonine-charged tRNAs to the 60*S* subunit (*Shen et al., 2015*). Also, our ability to temporally separate CAT tailing from canonical translation in vitro (by the addition of exogenous Rqc2p to *rqc2Δ* extracts) provided an experimental strategy for specifically testing the requirements of CAT-tail elongation.

## Mechanistic differences between CAT-tail synthesis and canonical translation

Though previous structural studies of the RQC were instrumental in discovering CAT tailing and suggested a direct role for Rqc2p in the process (*Shen et al., 2015*), the mechanism of CAT tailing has only been inferred from these data and otherwise remains poorly characterized. In particular, no published studies have investigated the extent to which CAT tailing by the 60S subunit is mechanistically similar to canonical elongation by the 80S ribosome. To address this question, we sought to examine the sensitivity of in vitro CAT tailing to a collection of well-characterized chemical inhibitors that target different sites of the ribosome or elongation factors (*Figure 2A*) and thereby interfere with canonical translation. Importantly, because canonical translation is required to generate the substrate for the RQC (i.e., a stalled ribosome at the post-splitting stage), it was necessary to temporally

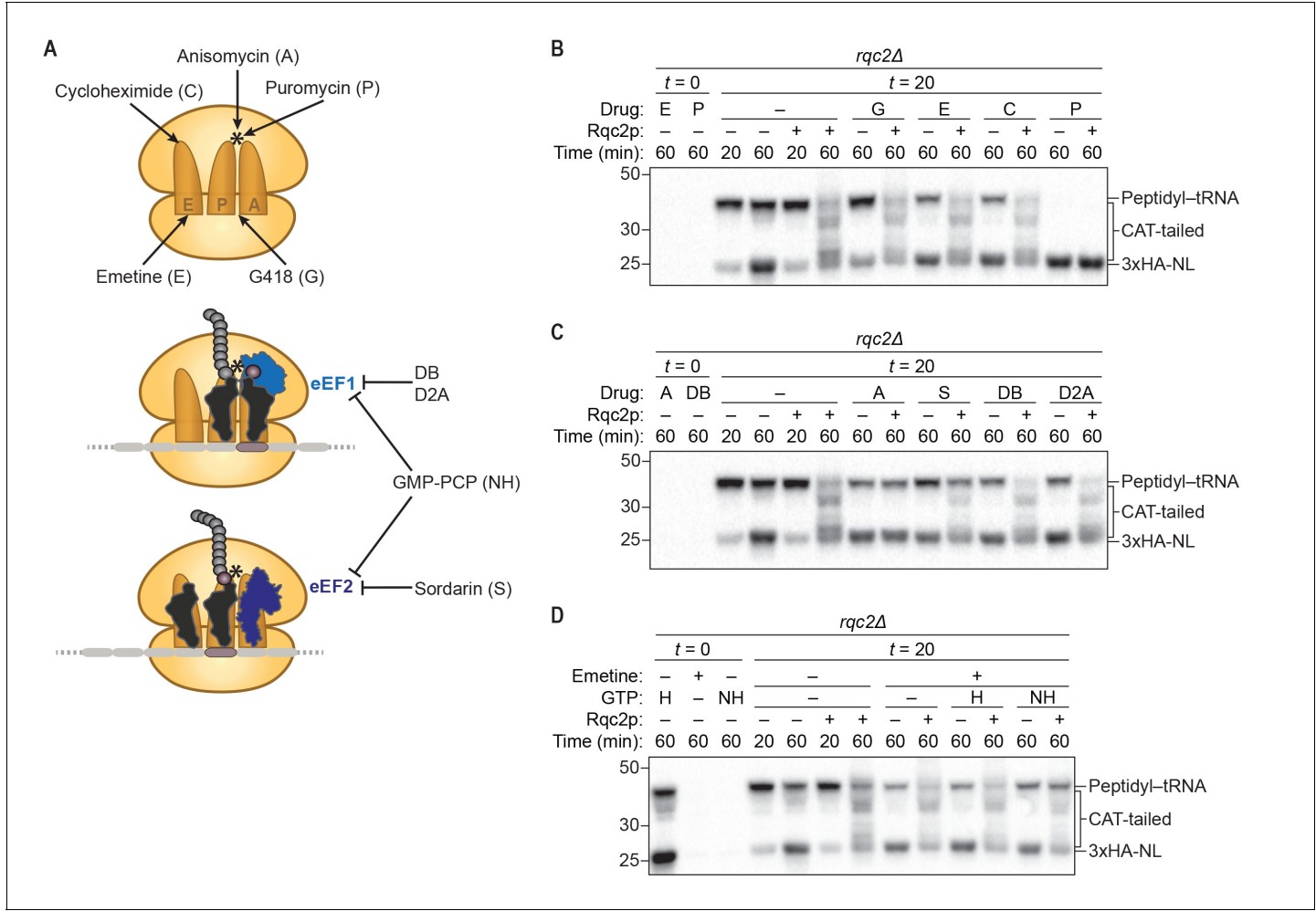

**Figure 2.** CAT-tail synthesis is mechanistically distinct from canonical translation. (**A**) Schematics of small-molecule inhibitors that directly bind the ribosome (top) or that target the translation elongation factors eEF1a or eEF2 (bottom). Inhibitors: (A) anisomycin; (C) cycloheximide; (D2A) didemnin 2A; (DB) didemnin B; (E) emetine; (G) G418; (H) hydrolyzable GTP; (NH) non-hydrolyzable GTP-analog GMP-PCP; (P) puromycin; (S) sordarin. (*) Denotes the peptidyl-transferase center of the 60S subunit. (**B–D**) Effects of small-molecule inhibitors on CAT tailing. ScIVT reactions were prepared using *rqc2Δ* extracts and a lysine-free truncated mRNA substrate. After 0 min (*t* = 0) or 20 min (*t* = 20) of translation, reactions were supplemented with either protein storage buffer (–) or purified Rqc2p at 670 nM final concentration (+) and the indicated inhibitor(s). Indicated time points ('Time (min)') were analyzed by SDS-PAGE and immunoblotting. Additional *t* = 0 controls for the remaining inhibitors can be found in *Figure 2—figure supplement 1*.

The following figure supplement is available for figure 2:

**Figure supplement 1.** Inhibitors and purified Rqc2p used to dissect the mechanism of CAT-tail synthesis.

separate canonical translation from CAT tailing to isolate the effects of these inhibitors on the latter reaction. To do so, we first translated a truncated mRNA in *rqc2Δ* extracts in the absence of any inhibitors for 20 min to generate a pool of RQC substrate (60*S*:peptidyl–tRNA). We then supplemented the reactions with one of the inhibitors (at a concentration that completely inhibited canonical translation in vitro; *Figure 2—figure supplement 1A*) and purified Rqc2p (*Figure 2—figure supplement 1B*) (or a buffer-only control) to initiate CAT-tail synthesis. Finally, we allowed the reactions to proceed for an additional 40 min before analyzing the products by immunoblotting.

As predicted by the structural analyses, treatment with drugs that target the peptidyl-transferase center (PTC) of the 60*S* subunit (anisomycin and the chain terminator puromycin) completely prevented CAT-tail synthesis, providing direct evidence that the catalytic activity of the ribosome is required for CAT tailing (*Figure 2B and C*). Puromycin treatment also resulted in a complete collapse of the 43 kDa mass-shifted species to 23 kDa, confirming the identity of this larger species as peptidyl–tRNA (*Figure 2B*). In contrast to the dramatic effects of PTC inhibitors, inhibitors that target the 40*S* subunit (emetine and G418) did not inhibit CAT-tail synthesis (*Figure 2B*), consistent with the proposed 40*S* subunit–independent mechanism of CAT tailing. Surprisingly, cycloheximide—which binds in the E site of the 60*S* subunit and sterically clashes with the 3′ end of the deacylated tRNA during canonical translation—did not inhibit CAT tailing (*Figure 2B*). Cycloheximide insensitivity identifies an unanticipated feature of CAT-tail elongation that may reflect a distinct mechanism of deacylated-tRNA displacement in the absence of mRNA and the 40*S* subunit.

It was previously proposed that specific Rqc2p–tRNA interactions impart selectivity for alanine- and threonine-tRNAs to CAT-tail elongation (*Shen et al., 2015*). However, it is not known if the translation elongation factors eEF1a and eEF2—which deliver aminoacyl-tRNAs to the ribosome and promote translocation, respectively—collaborate with Rqc2p to facilitate CAT-tail synthesis. Strikingly, we found that drugs targeting either eEF1a or eEF2 (didemnin variants (*Carelli et al., 2015*) or sordarin (*Justice et al., 1998*), respectively) had no effect on CAT tailing (*Figure 2C*). Because many canonical translation factors are GTPases, including eEF1a and eEF2, we also examined whether CAT tailing requires GTP hydrolysis. We tested for inhibition by the non-hydrolyzable GTP analog GMP-PCP, using a similar approach as before except that at the time of Rqc2p addition we stopped translation by adding emetine to prevent ongoing translation in the GTP control reaction. Consistent with the above differences between CAT-tail elongation and translation, treatment with GMP-PCP had no impact on CAT tailing (*Figure 2D*), indicating that this mRNA-independent elongation mechanism does not require energy from GTP hydrolysis or the canonical activities of the translational GTPases eEF1a and eEF2. Collectively, these findings provide direct evidence that CAT tailing is a 40*S* subunit–independent, PTC-catalyzed reaction and identify key differences from canonical translation that suggest an entirely different elongation cycle.

## Ltn1p- and Rqc1p-dependent ubiquitination in the yeast cell-free system

The ability of ScIVT to recapitulate Rqc2p-dependent CAT tailing (*Figure 1*) led us to explore whether this system also recapitulates the other key activity of the RQC, Ltn1p-dependent nascent-chain ubiquitination. Initial experiments comparing a lysine-containing truncated reporter mRNA to a lysine-free version revealed a faint smear of high-molecular-weight (HMW) products (~50–115 kDa) unique to the lysine-containing reporter (*Figure 3A*, compare lanes 1 and 4). We reasoned that these HMW products were likely ubiquitinated proteins because lysine residues are the canonical ubiquitination sites (*Pickart, 2001*). However, because ScIVT extracts contain many ubiquitin ligase activities and a finite pool of endogenous ubiquitin, we suspected that Ltn1p-dependent ubiquitination of the reporter protein (which occurs late in the reactions) might have been limited by the amount of ubiquitin available to Ltn1p. Indeed, supplementing ScIVT reactions with exogenous ubiquitin resulted in enhanced accumulation of the lysine-dependent HMW products (*Figure 3A*). Treatment of ubiquitin-supplemented reactions with TEV protease or RNase did not fully collapse the HMW species as it did for CAT-tailed species (*Figure 3B*), consistent with the HMW species containing ubiquitin-modified residues rather than simply having exceptionally long CAT tails. To directly demonstrate that these HMW species contained ubiquitin, we translated truncated mRNAs (with or without lysines) in the presence of exogenous Myc-tagged ubiquitin and purified the reporter protein under denaturing conditions, followed by immunoblotting to detect Myc-tagged ubiquitin. In

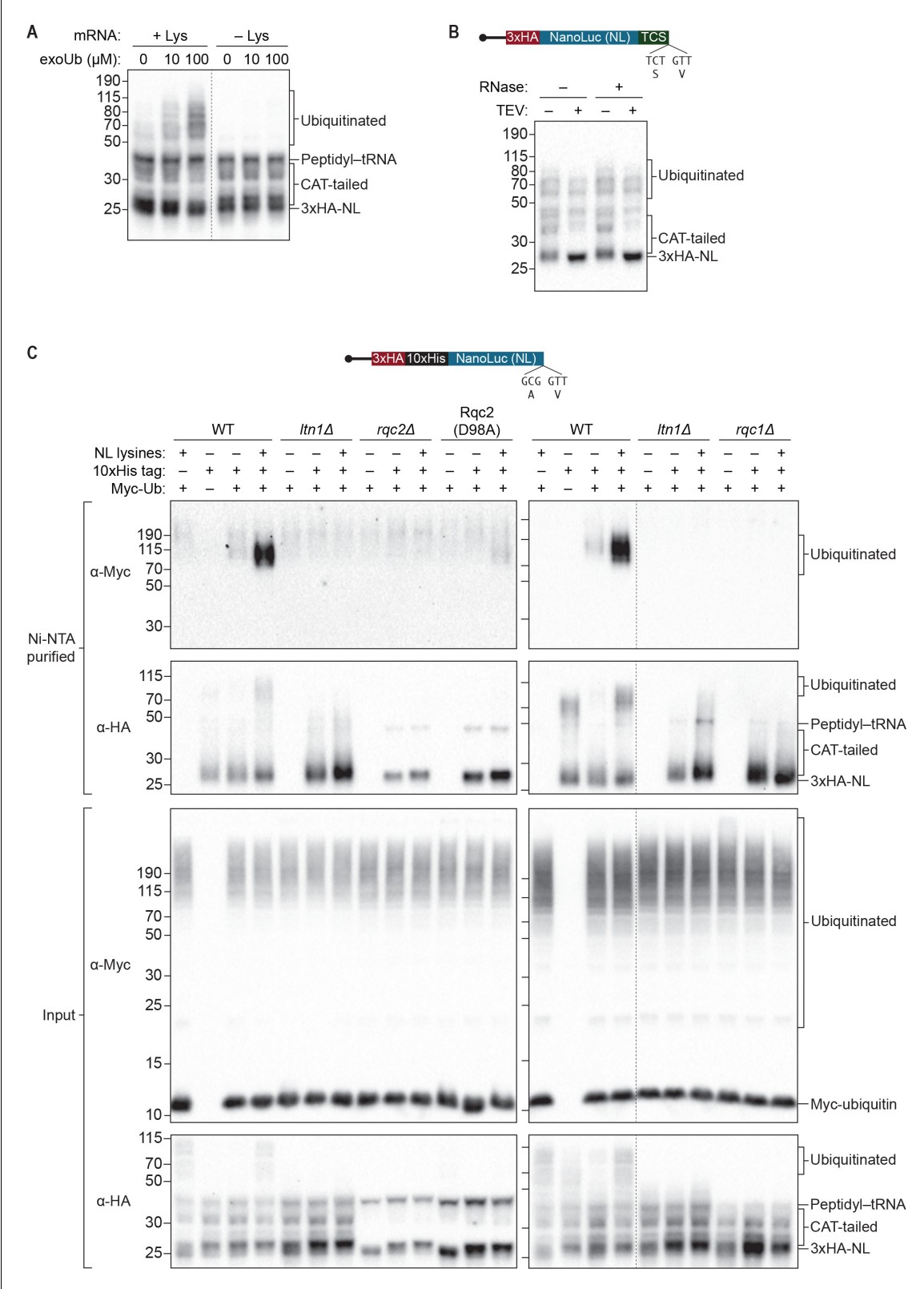

**Figure 3.** *S. cerevisiae* in vitro translation recapitulates Ltn1p-mediated ubiquitination. (**A**) Effects of adding exogenous ubiquitin to ScIVT reactions. ScIVT reactions conducted in WT extracts with lysine-containing (+Lys) or lysine-free (–Lys) truncated mRNA were supplemented with the indicated concentrations of recombinant ubiquitin, incubated for 60 min, and then analyzed by SDS-PAGE and immunoblotting. Dashed line indicates where intervening lanes were removed for clarity. (**B**) Analysis of high-molecular-weight smears. RNase A/T1 and TEV protease treatment of ScIVT reactions

*Figure 3 continued on next page*

Figure 3 continued

programmed with lysine-containing truncated mRNA encoding a TEV cleavage site (TCS) in WT extracts supplemented with 100 µM recombinant ubiquitin. Translation was halted after 60 min by addition of 20 mM EDTA, after which reactions were treated without (–) or with (+) TEV and/or RNase A/T1 for 60 min, and then analyzed by SDS-PAGE and immunoblotting. Note that due to long incubations (120 mins), very little peptidyl–tRNA persists in these reactions. (C) Isolation and detection of ubiquitinated ScIVT products. ScIVT reactions were conducted with 1.2 µg of truncated mRNA (3xHA-10xHis-NanoLuc, with or without lysines or His tag as indicated) in extracts prepared from strains of the indicated genotypes and supplemented with 10 µM recombinant Myc-ubiquitin. For input samples (bottom panels), one-third of the ScIVT reaction was quenched with 2X Laemmli Sample Buffer. For Ni-NTA-purified samples (top panels), two-thirds of the ScIVT reaction was quenched with 6 M guanidine-HCl. For SDS-PAGE, 30% of input samples and 100% of Ni-NTA-purified samples were separated on 12% NuPAGE gels and translation products were visualized by immunoblotting with antibodies indicated at left. Dashed lines indicate where intervening lanes were removed for clarity.

reactions conducted with WT extracts, we readily detected Myc-tagged ubiquitin within the purified HMW species (*Figure 3C*, fourth lanes in left and right panels).

As expected, we did not observe ubiquitination of the reporter protein in reactions performed with *ltn1Δ* extracts (*Figure 3C* and *Figure 4A*), consistent with Ltn1p being the responsible E3 ubiquitin ligase. The addition of purified Ltn1p to *ltn1Δ* extracts restored ubiquitination, while the addition of purified Ltn1p containing the W1542E substitution—a RING domain mutant that does not support protein turnover in vivo (*Bengtson and Joazeiro, 2010*)—did not (*Figure 4A* and *Figure 4— figure supplement 1*). These results demonstrate that the lack of ubiquitination in *ltn1Δ* extracts is a direct consequence of the absence of Ltn1p rather than an indirect effect. Collectively, these results demonstrate that in addition to Rqc2p-dependent CAT tailing the ScIVT system we established also recapitulates Ltn1p-dependent ubiquitination, as previously shown for lysates derived from *N. crassa* and rabbit reticulocytes (*Doamekpor et al., 2016*; *Shao et al., 2013*).

Unexpectedly, we did not detect any Ltn1p-dependent ubiquitination of the reporter protein in *rqc1Δ* extracts (*Figure 3C*). The addition of purified Rqc1p (*Figure 4—figure supplement 1*), however, fully rescued ubiquitination in *rqc1Δ* extracts (*Figure 3C* and *Figure 4B*). Increasing the concentration of Ltn1p in the reaction did not bypass the requirement for Rqc1p in ubiquitination (*Figure 4—figure supplement 2*). These observations suggest that Rqc1p is directly involved in nascent-chain ubiquitination. Such a role is consistent with the fact that *LTN1* deletion phenocopies *RQC1* deletion in both the accumulation of stalling reporters in vivo and, more broadly, in their correlated set of genetic interactions (*Brandman et al., 2012*; *Defenouillère et al., 2013*). Although TCF25/Rqc1p was previously reported to be dispensable for Listerin/Ltn1p-mediated ubiquitination of purified 60S-bound stalled nascent chains (*Shao and Hegde, 2014*), the stringent purification of ribosome–nascent chain complexes in that study might have removed factors that otherwise impose a requirement for Rqc1p/TCF25 (e.g., chaperones that protect the nascent chain). Our results support a model in which Rqc1p directly promotes Ltn1p-mediated ubiquitination of the nascent chain via a mechanism that remains to be determined.

## Interplay between CAT tailing and ubiquitination

Previous studies have shown that a CAT-tailing-deficient mutant of Rqc2p preserves degradation of stalled nascent chains (*Shen et al., 2015*) and that in vitro reconstitution of Listerin/Ltn1p-mediated ubiquitination does not strictly require NEMF/Rqc2p (*Shao and Hegde, 2014*; *Shao et al., 2013*). Together, these studies suggested that CAT tailing is dispensable for degradation of the assayed reporter constructs. Based on these results and structural data, it was proposed that Rqc2p indirectly contributes to ubiquitination by recognizing the aberrant 60S:peptidyl–tRNA complex, stabilizing Ltn1p on the 60S subunit, and sterically preventing the 40S subunit from rejoining (*Shao et al., 2015*). Together with the minimal impact of *LTN1* disruption on CAT tailing in vivo (*Choe et al., 2016*; *Defenouillère et al., 2016*; *Shen et al., 2015*; *Yonashiro et al., 2016*) and in vitro (*Figure 1*), these studies led to a model in which ubiquitination by Ltn1p and CAT tailing by Rqc2p are independent activities of the RQC (*Inada, 2017*).

The unique ability of our ScIVT system to recapitulate both activities of the RQC, combined with its genetic tractability, allowed us to directly test this model in vitro. Consistent with the proposed scaffolding function of Rqc2p in ubiquitination, disruption of *RQC2* abrogated nascent-chain ubiquitination by Ltn1p (*Figure 3C* and *Figure 4C*), and addition of wild-type Rqc2p rescued ubiquitination

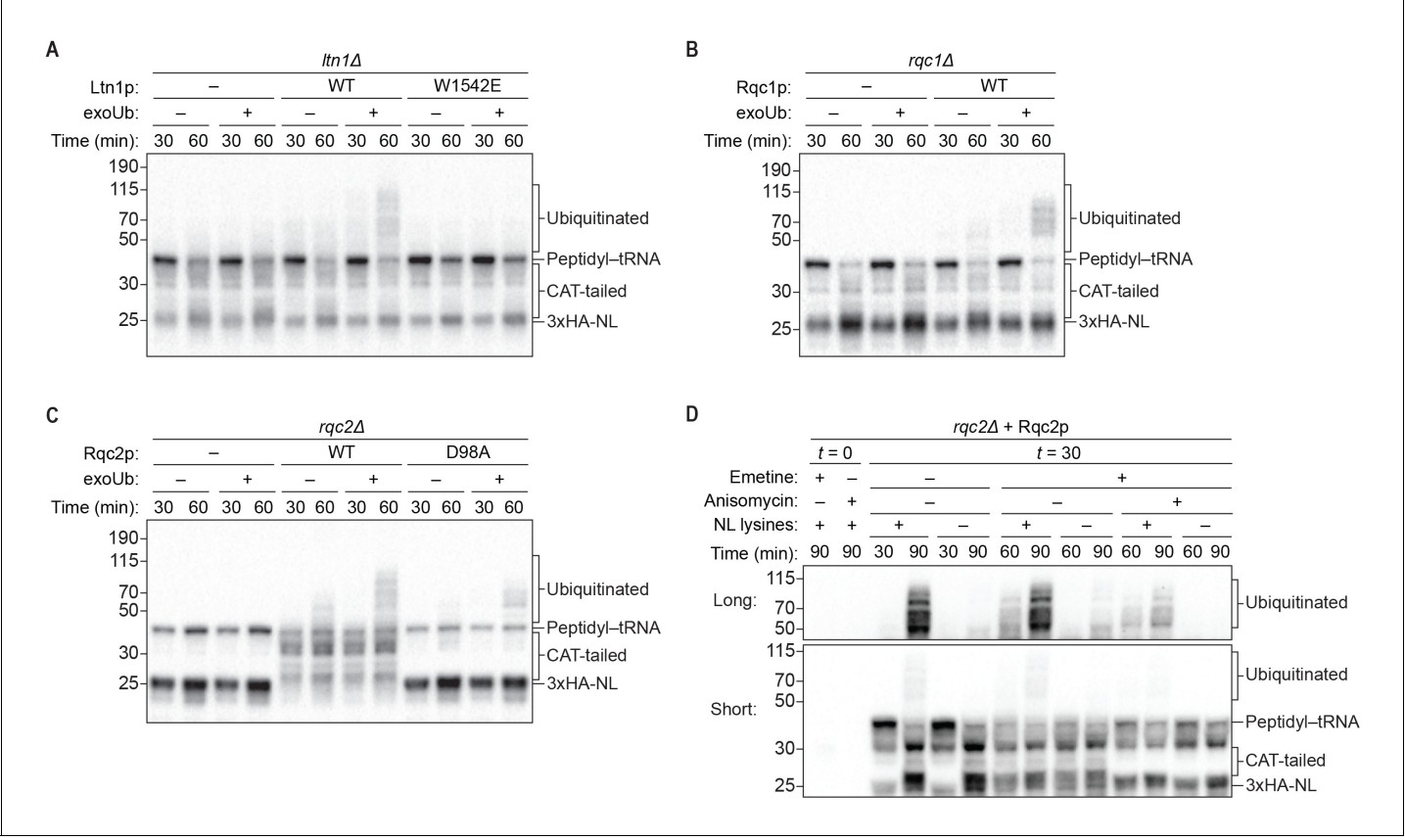

**Figure 4.** Rqc1p and CAT tailing contribute to Ltn1p-dependent ubiquitination. (**A–C**) Genetic analysis of RQC-mediated ubiquitination in ScIVT. ScIVT reactions were prepared using extracts from strains of the indicated genotype, a lysine-containing truncated mRNA substrate, ubiquitin storage buffer (–) or 100 μM recombinant ubiquitin (+), and either protein storage buffer (–) or the indicated purified proteins (+): Ltn1p at 130 nM, Rqc1p at 70 nM, and Rqc2p at 420 nM final concentration. (**D**) ScIVT reactions were conducted using *rqc2Δ* extracts, a lysine-free or lysine-containing truncated mRNA substrate, and 100 μM exogenous ubiquitin. After 0 min (*t* = 0) or 30 min (*t* = 30) of translation, all reactions were supplemented with an equal volume of 'mock ScIVT' (i.e., without mRNA) containing 1.34 μM purified Rqc2p, 100 μM exogenous ubiquitin, and the indicated inhibitor(s). Indicated time points ('Time (min)') were analyzed by SDS-PAGE and immunoblotting. 'Long' and 'Short' refer to exposure times of the blots.

The following figure supplements are available for figure 4:

**Figure supplement 1.** Purified Ltn1p and Rqc1p.

**Figure supplement 2.** Impact of excess Ltn1p on ubiquitination in *rqc1Δ* extracts.

**Figure supplement 3.** Impact of CAT-tailing inhibition on ubiquitination.

(*Figure 4C* and *Figure 1—figure supplement 1*). Unexpectedly, however, the addition of CAT-tailing-deficient (D98A) Rqc2p to *rqc2Δ* extracts only partially rescued ubiquitination (*Figure 4C*). Similarly, reactions using extracts containing endogenously expressed Rqc2p(D98A) as the only *RQC2* gene product yielded minimal ubiquitinated protein (*Figure 3C*). To rule out the possibility that the effect of the D98A substitution on ubiquitination was due to disruption of the known scaffolding function of Rqc2p, we took an alternative approach to inhibit CAT tailing in the context of wild-type Rqc2p. As observed in the Rqc2p(D98A) experiments, preventing CAT tailing of stalled nascent chains—in this case with anisomycin treatment (*Figure 2C*)—substantially reduced ubiquitination (*Figure 4D* and *Figure 4—figure supplement 3*). These results suggest that Rqc2p not only provides structural support for Ltn1p but also that CAT tailing directly enhances Ltn1p-dependent ubiquitination of at least some substrates (see Discussion). Collectively, our in vitro analyses reveal that

all three components of the RQC—Ltn1p/Listerin, Rqc1p/TCF25, and Rqc2p/NEMF—contribute to ubiquitination of the nascent chain.

## Discussion

We have shown that establishing a cell-free system that recapitulates both CAT tailing and ubiquitination opens new opportunities to explore how the fully functional RQC promotes clearance of aberrant translation products. Our analyses reveal that Rqc2p-mediated nascent-chain elongation is mechanistically distinct from canonical translation, that ubiquitination of the nascent polypeptide requires both Ltn1p and Rqc1p, and that the ubiquitination and CAT-tailing activities of the RQC are coupled through a mutual requirement for active Rqc2p (*Figure 5*).

The key benefit of an in vitro system to study CAT tailing is the ability to perform experiments that would be intractable in vivo. Indeed, a major difficulty in studying CAT tailing is that it utilizes some of the same machinery as canonical translation (i.e., the 60S subunit) and requires a substrate that is generated by canonical translation, making it difficult to perturb CAT tailing specifically. By temporally separating the translation- and CAT-tailing-phases of in vitro reactions, we overcame this obstacle and specifically tested the sensitivity of the CAT-tailing reaction to a wide range of well-

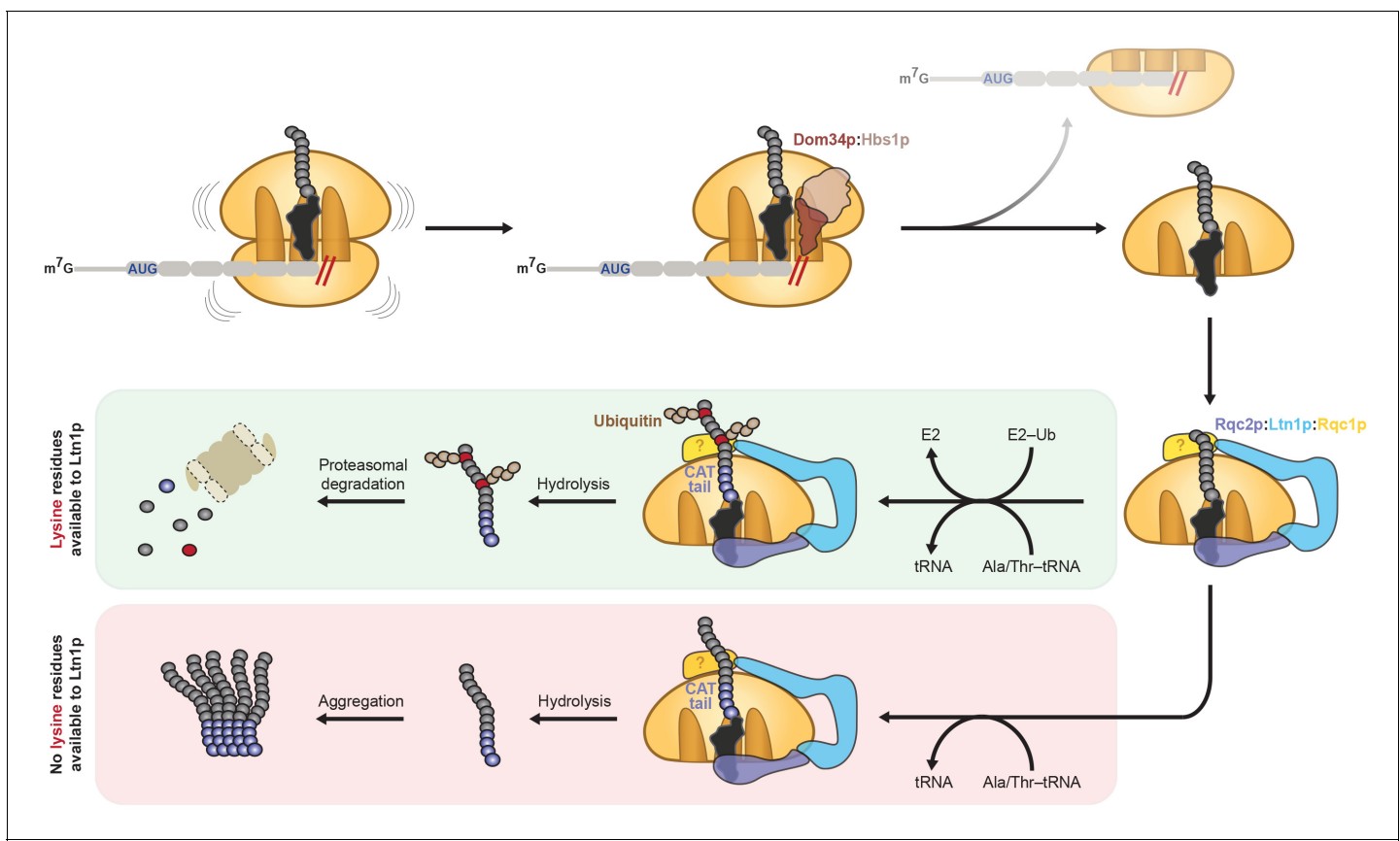

**Figure 5.** Model for CAT tailing and ubiquitination of stalled nascent chains. When an 80S ribosome stalls during translation, splitting factors recognize the stalled translation complex to facilitate dissociation of the 40S subunit and mRNA. Ltn1p, Rqc2p, and Rqc1p (unknown location indicated by '?') bind the resulting 60S:peptidyl–tRNA complex. Together with the peptidyl-transferase center of the 60S subunit, Rqc2p facilitates elongation of the stalled nascent chain with a CAT tail by recruiting alanine- and threonine-charged tRNAs to the A site. If the nascent chain contains a lysine residue (red circle) located within the vicinity of the Ltn1p RING domain (or potentially hidden inside the ribosome exit tunnel), CAT tailing and Rqc1p enhance or facilitate Ltn1p-mediated ubiquitination of the nascent chain, respectively, for subsequent proteasomal degradation (green box). If the nascent chain does not contain any lysine residues (or contains lysine residues that are too distant from the Ltn1p RING domain), CAT tails may promote aggregation of incompletely synthesized proteins (red box). In both instances, Rqc2p activity promotes hydrolysis of the peptidyl–tRNA linkage and liberation from the 60S subunit.

characterized ribosome and translation inhibitors. These analyses revealed that aside from requiring the catalytic PTC of the ribosome, CAT tailing is otherwise fundamentally different from translation elongation in ways that could not have been fully anticipated from previous studies (*Figure 2*). In particular, our findings that CAT tailing does not require the canonical activities of the elongation factors or energy from GTP hydrolysis suggest a unique mechanism of elongation. Reexamining 60*S*: RQC structures (*Shao et al., 2015*; *Shen et al., 2015*), we noted that the network of interactions between Ltn1p/Rqc2p and the 60*S* subunit overlaps with the ribosome-binding sites of eEF1a (*Shao et al., 2016*) and eEF2 (*Taylor et al., 2007*). The overlap in binding sites indicates that either eEF1a/eEF2 are dispensable for CAT tailing, or that eEF1a/eEF2 and the RQC interact transiently with the 60*S* subunit during elongation. Our findings favor the former model and suggest that Rqc2p directly recruits charged tRNAs with its selective tRNA-binding activity (*Shen et al., 2015*), without the involvement of eEF1a. Following peptide-bond formation by the 60*S* subunit, the A/P and P/E tRNAs may spontaneously translocate in the absence of interactions with an mRNA template or 40*S* subunit; whereas during canonical translation such interactions impose an energy requirement for translocation that is fulfilled by the GTPase activity of eEF2. Given that CAT tailing was proposed to occur on the 60*S* subunit (*Shen et al., 2015*), we predicted that all translation inhibitors that bind the 60*S* subunit would inhibit CAT-tail synthesis. However, we found that CAT tailing was not inhibited by cycloheximide, suggesting that the deacylated tRNA may rapidly dissociate from the 60*S* subunit following peptidyl transfer.

Together, these findings suggest that a minimal set of factors—a 60*S*:peptidyl–tRNA complex, charged alanine and threonine tRNAs, and Rqc2p—may be sufficient for CAT-tail synthesis. In 1969, Monro demonstrated that in the presence of certain alcohols, isolated bacterial 50*S* ribosomal subunits could catalyze polymerization from aminoacyl-tRNAs in the absence of an mRNA template and 30*S* subunits (*Monro, 1969*). Thus, it is conceivable that Rqc2p may stimulate nascent-chain elongation much as in Monro's minimal prokaryotic system. Many questions remain about Rqc2p dynamics during CAT tailing, including whether a single molecule of Rqc2p remains on the 60*S* subunit for successive cycles of peptide-bond formation or whether each cycle of elongation requires binding by a new Rqc2p–tRNA complex.

CAT-tail synthesis eventually terminates by hydrolysis of the peptidyl–tRNA linkage, which is presumably needed to release the CAT-tailed nascent chain from the 60*S* subunit for its destruction by the proteasome. Indeed, one critical function of CAT tailing might be to provide a mechanism of termination in the absence of a stop codon. The wide range of CAT-tail lengths observed both in vivo (*Choe et al., 2016*; *Defenouillère et al., 2016*; *Shen et al., 2015*; *Yonashiro et al., 2016*) and in vitro suggests that termination is a stochastic process. Furthermore, our finding that peptidyl–tRNA intermediates are relatively stable in vitro in the absence of Rqc2p (*Figure 1D and F*, *Figure 2*, and *Figure 4C*) indicates a potentially direct role of Rqc2p in the termination reaction. These observations lead to a model for CAT-tailing termination in which Rqc2p recruits a termination factor to the 60*S* subunit in a stochastic manner during the process of elongation. Because Rqc2p interacts with A-site tRNAs at a site distant from the acceptor stem (*Shen et al., 2015*), it is possible that the 'termination factor' is an uncharged alanine- or threonine-tRNA (*Caskey et al., 1971*; *Zavialov et al., 2002*). Alternatively, Rqc2p might interact with the canonical termination factor eRF1 or another protein with peptidyl–tRNA hydrolase activity to facilitate termination.

Taking advantage of the fact that our extracts also recapitulated Ltn1p-dependent ubiquitination, we found that Rqc1p plays a critical role in nascent-chain ubiquitination in vitro (*Figure 3C* and *Figure 4B*). This direct role of Rqc1p in ubiquitination contrasts with its previously suggested role in recruiting Cdc48p downstream of ubiquitination (*Brandman et al., 2012*; *Defenouillère et al., 2013*). Nevertheless, our discovery that ubiquitination in vitro is as dependent on Rqc1p as it is on the E3 ligase Ltn1p is consistent with the fact that yeast strains lacking Rqc1p and Ltn1p have very similar phenotypes and genetic interaction profiles, suggesting a similar molecular defect in these strains (*Brandman et al., 2012*). We speculate that Rqc1p may facilitate ubiquitination by positioning the nascent chain in proximity to the Ltn1p RING domain, by promoting binding of the E2 ubiquitin-conjugating enzyme, or by activating Ltn1p's E3 ligase activity on the 60*S* subunit.

With a system that uniquely recapitulates both nascent-chain elongation by Rqc2p and ubiquitination by Ltn1p, we discovered that Rqc2p can elongate the nascent chain to enhance ubiquitination, rather than just providing structural support for Ltn1p binding to the 60*S* subunit (*Figure 3C* and *Figure 4C*). This finding was surprising given that previous studies have shown that a CAT-tailing-

deficient mutant of Rqc2p preserves degradation of aberrant nascent chains in yeast cells (*Shen et al., 2015*) and that in vitro reconstitution of Listerin/Ltn1p-mediated ubiquitination does not strictly require NEMF/Rqc2p (*Shao and Hegde, 2014*). How can we reconcile these findings? Structural studies of full-length Listerin/Ltn1p on the 60S subunit localized its RING domain (which binds an E2 ubiquitin-conjugating enzyme) near the ribosomal exit tunnel (*Shao et al., 2015*), poised to facilitate ubiquitination of lysine residues close to or recently emerged from the tunnel. The physical tethering of the RING domain near the exit tunnel suggests that Ltn1p may not be able to access more distantly positioned lysines in the nascent polypeptide, nor can the most recently translated lysines—contained within the 30–60-amino-acid long exit tunnel (*Kramer et al., 2009*)—be accessed by Ltn1p until their emergence. Our observations lead to a model in which Ltn1p can only ubiquitinate a spatially restricted set of lysines, while CAT tailing enables access to other lysines—as previously proposed (*Brandman and Hegde, 2016*; *Simms et al., 2017*) and recently demonstrated in vivo (*Kostova KK et al., 2017*). We reason that the few nascent chain RQC substrates previously studied in vivo could be degraded in a CAT-tailing-independent manner (*Choe et al., 2016*; *Defenouillère et al., 2016*; *Shen et al., 2015*; *Yonashiro et al., 2016*) due to native lysines being fortuitously positioned proximal to the Ltn1p RING domain.

Recent studies in budding yeast have demonstrated that CAT tails mediate formation of detergent-insoluble aggregates when the nascent chain cannot be degraded due to its limited ubiquitination potential or due to inactivation of Ltn1p (*Choe et al., 2016*; *Defenouillère et al., 2016*; *Yonashiro et al., 2016*). In the context of a fully intact RQC, however, our findings suggest that CAT tailing and ubiquitination are interdependent activities. Elongation of the nascent chain with CAT tails can result in two outcomes: positioning lysine residues proximal to the Ltn1p RING domain for efficient ubiquitination; or distancing lysine residues from the Ltn1p RING domain, making ubiquitination less efficient. Thus, CAT tailing and ubiquitination must be tightly coordinated to promote nascent-chain degradation and to avoid, where possible, aggregate formation and the detrimental sequestration of cytosolic chaperones. These studies collectively suggest that rather than being the primary role of CAT tails, aggregation is more likely a backup pathway to mitigate the toxic effects of stalled polypeptides that cannot be efficiently ubiquitinated and degraded (*Figure 5*). A notable distinction between the aggregation- and ubiquitination-promoting functions of CAT tails is that the former depends on the alanine/threonine composition of the CAT tail (*Choe et al., 2016*), while the latter depends on the process of CAT tailing itself.

While CAT tailing has yet to be reported in metazoans, the conservation of Rqc1p/TCF25, Rqc2p/NEMF, and Ltn1p/Listerin—including critical residues that we mutated in this study—suggest that both RQC activities are conserved and together provide a means of protecting cells against the accumulation of faulty translation products. Ltn1p-dependent ubiquitination has been detected in rabbit reticulocyte extracts, so a CAT-tailing-dependent mechanism to facilitate ubiquitination of inaccessible lysine residues likely operates in metazoans as well. Underscoring the importance of this quality-control mechanism in maintaining proteostasis, mutations in the mammalian homologs of *HBS1* (*Ishimura et al., 2014*) and *LTN1* (*Chu et al., 2009*) cause neurodegeneration in mice.

With our newfound insights, the similarities between the bacterial tmRNA system and the eukaryotic RQC become even more striking than previously appreciated. In certain bacteria, a stalled ribosome is rescued by the recruitment of a hybrid tRNA/mRNA-like molecule (tmRNA) to the empty A-site of the ribosome, leading to translation of a tmRNA-encoded C-terminal degron that includes a dedicated stop codon (*Moore and Sauer, 2007*). In this way, stalled nascent chains are marked for degradation and translation can terminate even when the mRNA template lacks a stop codon. The RQC fulfills these same functions but in a different manner that is compatible with the ubiquitin-proteasome system: Stalled nascent chains are marked for degradation by ubiquitination (in certain cases facilitated by CAT tailing), and translation can terminate without a stop codon with the help of Rqc2p. Thus, both mechanisms involve a ribosome-catalyzed peptidyl-transferase reaction that adds a C-terminal extension that is not templated by the parent mRNA molecule. The addition of the C-terminal extension, moreover, facilitates peptidyl–tRNA hydrolysis and nascent-chain release either directly (tmRNA) or indirectly (RQC) to promote degradation. These functional similarities between the tmRNA system and the RQC—despite not sharing any related factors other than the ribosome—provide a striking example of convergent evolution that emphasizes the physiological importance of discarding incompletely synthesized proteins and recycling the translation machinery.

## Materials and methods

### Yeast strain construction

*Saccharomyces cerevisiae* strains used in this study were derived from BY4741 (*MATa his3Δ1 leu2Δ0 met15Δ0 ura3Δ0*) and are listed in *Supplementary file 1*. Yeast transformations were performed using the PEG–lithium acetate method (*Gietz and Woods, 2006*). To generate gene knockouts, the entire coding sequence of the gene of interest was replaced with the *URA3* cassette of pRS416. Strains containing point mutations at endogenous loci were generated from *URA3*-disrupted strains by transformation with PCR fragments encoding the mutant gene of interest and 5-FOA counterselection (*Boeke et al., 1987*). Transformants were screened by PCR to identify integrants, which were subsequently verified by PCR and Sanger sequencing of the entire integrated cassette.

### Preparation of mRNA for in vitro translation

pcDNA3.1(+) (Thermo Fisher Scientific, Waltham, MA) was modified using Gibson Assembly Master Mix (New England Biolabs, Ipswich, MA) and appropriate DNA fragments according to the Gibson assembly method (*Gibson et al., 2009*) to generate pBAO1124, which contains (in order): T7 promoter, 46-nt 5′-UTR lacking any AUG or near-AUG codons (i.e., NUG, ANG, or AUN, where N is any nucleotide), 3xHA-NanoLuc luciferase ORF, 56-nt 3′-UTR, and 50-nt poly(A) sequence. RNAs were generated by run-off transcription with T7 RNA Polymerase using the MEGAscript T7 Transcription Kit (Thermo Fisher Scientific) according to the manufacturer's instructions using PCR-amplified DNA templates derived from pBAO1124 or its variants (DNA sequences are provided in *Supplementary file 2*). Transcription reactions were terminated by addition of ammonium acetate stop solution. RNA was extracted with neutral phenol:chloroform:isoamyl alcohol (25:24:1) (Sigma, St. Louis, MO), precipitated with ethanol, and resuspended in nuclease-free water. A 5′−7-methyl-guanosine cap was added to RNA post-transcriptionally using the Vaccinia Capping System (New England Biolabs). Capping reactions were desalted using Micro Bio-Spin Columns with Bio-Gel P-30 (Bio-Rad, Hercules, CA) before RNA was extracted with phenol, precipitated with ethanol, and resuspended in nuclease-free water.

### *S. cerevisiae* in vitro translation extract preparation

*S. cerevisiae* strains were grown overnight to saturation in rich YPAD media, diluted the next morning to $OD_{600}$ 0.2 in a total volume of 1 L YPAD, and harvested at $OD_{600}$ 1.4–1.8 by centrifugation at 3500 rpm for 6 min at 4°C. The cell pellet was washed with water and resuspended in lysis buffer A (30 mM HEPES-KOH pH 7.4, 100 mM KOAc, 2 mM Mg(OAc)$_2$, 2 mM DTT, and cOmplete mini EDTA-free protease inhibitor cocktail [Roche, Switzerland]) using 1 ml per 6 g of wet cell pellet. The cell slurry was dripped into liquid nitrogen to produce frozen pellets, which were then pulverized using a 6970EFM Freezer/Mill (SPEX SamplePrep, Metuchen, NJ) by three cycles of 12 Hz agitation for 1.5 min with cooling for 2 min after each cycle. The resulting 'grindate' was combined with an equal volume of pre-chilled lysis buffer A (i.e., 1 ml per 1 g of grindate) and allowed to thaw on ice. Cell debris was cleared by sequential centrifugation at 4°C at 1000 g for 5 min, 1350 g for 5 min, 14000 g for 30 min, and finally 14000 g for an additional 10 min. The clarified lysate was dialyzed twice for 2 hr against 250 ml lysis buffer A (except without protease inhibitor cocktail) using 3500 MWCO cassettes (Thermo Fisher Scientific #87722). After dialysis, lysates were flash frozen in 50 µl aliquots in liquid nitrogen and stored at –80°C.

### *S. cerevisiae* in vitro translation

Endogenous mRNAs in thawed extracts were degraded by treatment with 0.3 U/µl micrococcal nuclease (MNase) and 480 µM CaCl$_2$ for 10 min at room temperature, followed by addition of 2 mM EGTA and transfer to ice. ScIVT reactions were initiated by adding m$^7$G-capped RNA (40 ng per µl of reaction volume) to MNase-treated yeast extracts and incubating at 25°C for up to 90 min. Final concentrations of reaction components were 48.67% (v/v) MNase-treated yeast extract, 22 mM HEPES-KOH (pH 7.4), 120 mM potassium acetate, 1.5 mM magnesium acetate, 0.75 mM ATP, 0.1 mM GTP, 0.04 mM each amino acid, 1.7 mM DTT, 25 mM creatine phosphate, 0.34 µg/µl creatine kinase, 0.14 U/µl SUPERase•In RNase Inhibitor (Thermo Fisher Scientific), and 0.16X cOmplete mini EDTA-free protease inhibitor cocktail (Roche). Where indicated, reactions also included 10 or 100

μM recombinant human ubiquitin or Myc-ubiquitin (Boston Biochem, Cambridge, MA). Reactions were halted by transferring to ice or by adding an equal volume of 2X Laemmli Sample Buffer (Bio-Rad). The results shown for all ScIVT experiments are representative of at least two technical replicates (i.e., experiments conducted with independently prepared reagents).

## Inhibitor concentrations

Concentrated stock solutions are diluted 1:10 into the ScIVT reactions. For example: 1 μl of 2 mM Anisomycin (in 5% DMSO) was added to a 10 μl ScIVT reaction. All stock solutions should be dissolved in water (or diluted with water) as ScIVT reactions cannot tolerate 100% DMSO or ethanol. Small molecules were used at the following final concentrations:

- (A) anisomycin: 200 μM [stock = 2 mM in 5% DMSO]
- (C) cycloheximide: 300 μM [stock = 3 mM in water]
- (D2A) didemnin 2A: 50 μM [stock = 500 μM in 5% DMSO]
- (DB) didemnin B: 50 μM [stock = 500 μM in 5% DMSO]
- (E) emetine: 400 μM [stock = 4 mM in water]
- (G) G418: 200 μM [stock = 2 mM in water]
- (H) hydrolyzable GTP: 500 μM (i.e. 5x excess of GTP in reaction) [stock = 5 mM in water]
- (NH) non-hydrolyzable GTP-analog GMP-PCP: 500 μM (i.e. 5x excess of GTP in reaction) [stock = 5 mM in water]
- (P) puromycin: 500 μM [stock = 5 mM in water]
- (S) sordarin: 10 μM [stock = 100 μM in water]

## Denaturing purification of ScIVT products

30 μl ScIVT reactions were assembled with 1.2 μg 3xHA-10xHis-NL mRNAs and 10 μM recombinant Myc-ubiquitin (Boston Biochem) and incubated at 25°C for 1 hr. For input samples, 10 μl was removed and quenched in an equal volume of 2X Laemmli Sample Buffer. For Ni-NTA-purified samples, the remaining 20 μl was quenched by addition of denaturing buffer (6 M guanidine-HCl, 50 mM Tris pH 7.8, 300 mM KCl, 10 mM imidazole, 0.1% NP-40, 5 mM $\beta$-mercaptoethanol [$\beta$ME]), and then incubated with 10 μl of pre-washed Ni-NTA Magnetic Agarose Beads (Qiagen, Germany) at 4°C overnight with end-over-end rotation. Beads were washed three times with wash buffer I (denaturing buffer except 500 mM KCl) and three times with wash buffer II (denaturing buffer except 50 mM KCl and no guanidine-HCl), each for 5 min at room temperature. Bound proteins were eluted from beads by adding 15 μl elution buffer (wash buffer II except 200 mM imidazole) and incubating at 22°C for 5 min with shaking at 1000 rpm. The elution step was repeated, and eluates were pooled and mixed with an equal volume of 2X Laemmli Sample Buffer. The results shown for all experiments that include a denaturing purification of ScIVT products are representative of at least three technical replicates (i.e., experiments conducted with independently prepared reagents).

## Immunoblotting

Protein samples were separated by SDS-PAGE using 12% Bolt Bis-Tris gels (Thermo Fisher Scientific) and transferred in 1X CAPS Buffer onto 0.22 micron PVDF membrane (Bio-Rad). Blots were probed with the following antibodies diluted 1:5000 in 1X TBS-T containing 5% nonfat dry milk: mouse anti-HA (RRID:AB_627809, Santa Cruz Biotechnology [Dallas, TX] sc-7392), rat anti-HA high sensitivity (RRID:AB_390918, Roche #11867423001), mouse anti-Myc (RRID:AB_331783, Cell Signaling Technology [Danvers, MA] #2276), HRP-conjugated goat anti-mouse IgG (RRID:AB_631736, Santa Cruz Biotechnology sc-2005), and HRP-conjugated goat anti-rat IgG (RRID:AB_631755, Santa Cruz Biotechnology sc-2032). Blots were developed using Clarity ECL Western Blotting Substrate (Bio-Rad), and chemiluminescence was detected on a ChemiDoc Imaging System (Bio-Rad).

## Protein purification

*S. cerevisiae* strain yRH101 (a gift from Stephen Bell, MIT) derived from ySC7 (*Chen et al., 2007*) containing a 2 μm P$_{GAL1}$-[protein]−10xHis plasmid (a gift from Bob Stroud, UCSF) was grown overnight in SC−His media containing 2% raffinose, diluted the next day, and grown for an additional night to early saturation. Protein expression was induced by adding an equal volume of Yeast-Peptone media containing 2% galactose, and cells were grown for 5 hr at 30°C. Cells were harvested by

centrifugation at 3500 rpm for 6 min at 4°C, and the cell pellet was washed with water before resuspending in Lysis Buffer (20 mM HEPES-KOH pH 7.4, 500 mM KCl, 20 mM imidazole; 2 mM $\beta$ME and cOmplete EDTA-free protease inhibitor cocktail [Roche] added just prior to use) at a ratio of 1 ml per gram of cell pellet. The resulting cell slurry was dripped into liquid nitrogen to produce frozen pellets, which were pulverized using a 6970EFM Freezer/Mill (SPEX SamplePrep) by three cycles of 12 Hz agitation for 1.67 min with 2 min cooling after each cycle. The resulting powder was briefly thawed before adding Lysis Buffer (1 mL per 1 g of powder) supplemented with additional protease inhibitors (292 µM Pepstatin, 8.4 mM Leupeptin, 1.23 mM Aprotinin, 1 mM Phenylmethylsulfonyl fluoride [PMSF]). Cell debris was cleared by sequential centrifugation at 14000 rpm at 4°C for 10 min and then 30 min, followed by sequential filtration through 2.7 and 1.6 µm Whatman GD/X filters (GE Healthcare Life Sciences, Marlborough, MA). His-tagged proteins were purified from lysate using Ni-NTA Sepharose beads (Qiagen) as follows. Beads (~1 ml 50% slurry per 1 L yeast culture) were washed with water and equilibrated in Lysis Buffer. Lysate was added to semi-dry beads and rotated at 4°C for 2 hr. Beads were collected by centrifugation at 800 rpm for 3 min, resuspended in an equal volume of Lysis Buffer, loaded over a disposable Bio-Spin column (Bio-Rad), and washed once with 10 ml Lysis Buffer. The column was then washed as follows: once with 10 ml Wash Buffer (20 mM HEPES-KOH pH 7.4, 500 mM KCl, 10% glycerol, 2 mM $\beta$ME) containing 20 mM imidazole and 1 mM PMSF; once with 10 ml Wash Buffer containing 20 mM imidazole; and twice with 10 ml Wash Buffer containing 50 mM imidazole. Proteins were sequentially eluted from the beads by gravity rinses as follows: once with 250 µl Wash Buffer containing 250 mM imidazole; twice with 500 µl Wash Buffer containing 250 mM imidazole; and twice with 500 µl Wash Buffer containing 500 mM imidazole. Elution fractions were analyzed by SDS-PAGE and staining with Coomassie Brilliant Blue R-250 to identify those containing the protein of interest, which were then pooled and concentrated to ~500 µl before overnight dialysis into Rqc2p/Ltn1p Storage Buffer (50 mM HEPES-KOH pH 7.4, 300 mM KOAc, 5% glycerol, 2 mM DTT) or Rqc1p Storage Buffer (20 mM HEPES-KOH pH 7.4, 500 mM KOAc, 10% glycerol, 2 mM DTT). Dialyzed protein was passed through a 0.1 µm centrifugal filter (EMD Millipore [Hayward, CA] #UFC30VV00) before flash freezing in liquid nitrogen. Protein concentration was determined by both spectrophotometry using a Nanodrop 2000 (Thermo Fisher Scientific) and Coomassie staining against BSA standards.

## Acknowledgements

We thank Soufiane Aboulhouda for initial work on yeast in vitro translation and David Morgan and Peter Shen for comments on the manuscript. We also thank members of the Weinberg and Frost labs, Wendy Gilbert, Jonathan Weissman, and Kamena Kostova for insightful discussions, Jonathan Asfaha for experimental advice and reagents, and Jordan Carelli and Jack Taunton for eEF1a inhibitors. We also acknowledge Raul Andino, Keith Yamamoto, Alan Frankel, and David Agard for generously sharing equipment. This work was supported by an NSF Graduate Research Fellowship and a Moritz-Heyman Discovery Fellowship (BAO), a Hillblom Graduate Research Fellowship (CJH), the Searle Scholars Program 13SSP218 (AF), NIH grant 1DP2GM110772-01 (AF), an HHMI Faculty Scholar award (AF), the UCSF Program for Breakthrough Biomedical Research funded in part by the Sandler Foundation (AF, DEW), and an NIH Director's Early Independence Award DP5OD017895 (DEW). AF is a Chan Zuckerberg Biohub Investigator.

## Additional information

### Funding

| Funder | Grant reference number | Author |
|---|---|---|
| National Science Foundation | Graduate Research Fellowship | Beatriz A Osuna |
| UCSF Mortiz-Heyman Discovery Fellowship | Graduate Student Research Fellowship | Beatriz A Osuna |
| UCSF Hillblom Fellowship | Graduate Student Research Fellowship | Conor J Howard |

| Searle Scholars Program | 13SSP218 | Adam Frost |
|---|---|---|
| NIH Office of the Director | DP2GM110772 | Adam Frost |
| UCSF Program for Break-through Biomedical Research funded in part by the Sandler Foundation | | Adam Frost<br>David E Weinberg |
| NIH Office of the Director | DP5OD017895 | David E Weinberg |

The funders had no role in study design, data collection and interpretation, or the decision to submit the work for publication.

### Author contributions

BAO, Conceptualization, Investigation, Writing—original draft, Writing—review and editing; CJH, Investigation, Writing—review and editing; SK, Investigation; AF, Conceptualization, Supervision, Funding acquisition, Writing—review and editing; DEW, Conceptualization, Supervision, Funding acquisition, Writing—original draft, Writing—review and editing

### Author ORCIDs

Beatriz A Osuna, http://orcid.org/0000-0003-2604-6173
Conor J Howard, http://orcid.org/0000-0001-5375-6248
Adam Frost, http://orcid.org/0000-0003-2231-2577
David E Weinberg, http://orcid.org/0000-0002-9348-1709

## Additional files

### Supplementary files

• Supplementary file 1. *S. cerevisiae* strains used in this study. The names, genetic backgrounds, and protein expression plasmids of the strains generated and used in this study are listed in this table.

• Supplementary file 2. DNA sequences of the constructs used in this study.

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
