## [Decision Letter]

Thank you for submitting your article "*in vitro* analysis of RQC activities provides insights into the mechanism and function of CAT tailing" for consideration by *eLife*. Your article has been reviewed by three peer reviewers, one of whom is a member of our Board of Reviewing Editors and the evaluation has been overseen by a James Manley as the Senior Editor. The following individuals involved in review of your submission have agreed to reveal their identity: Alexander Mankin (Reviewer #2).

The reviewers have discussed the reviews with one another and the Reviewing Editor has drafted this decision to help you prepare a revised submission.

Summary:

This study reconstitutes in yeast extracts for the first time both the CAT-tailing and ubiquitylation activities of the RQC and exploits translation elongation inhibitors and mutants lacking different RQC components to probe the mechanisms and interdependencies between these two reactions. At odds with previously published work, they find that Rqc1 is required for ubiquitylation of the stalled peptides generated in their system using a non-STOP mRNA template, being defective in the rqc1Δ extract and rescued with purified Rqc1. As expected from the literature, CAT-tailing is defective in the rqc2Δ extract and can be rescued with purified WT Rqc2, but not with the D98A mutant. Unexpectedly, however, they find that the Rqc2-D98A substitution, in addition to deletion of the protein, impairs ubiquitylation as well as CAT-tailing, whereas previous studies had indicated no impact of this mutation on degradation of aberrant nascent chains in yeast cells, nor a requirement for Rqc2 in ubiquitylation of nascent chains in a mammalian extract. The authors propose instead that CAT-tailing and ubiquitylation are interdependent activities. They are also able to probe the effects of elongation inhibitors on the mechanism of CAT-tailing by the RQC complex using the preassembled, arrested 60S-peptidyl-tRNA complexes and report no effect of 40S inhibitors, as expected; nor of cycloheximide, which suggests no role for E-site binding of deacylated tRNA in CAT-tailing. They conclude that Inhibitors of the translation GTPases also had no effect, eliminating requirements for EF1 and EF2 in recruitment of charged Ala or Thr tRNAs and translocation, respectively; whereas inhibitors of the peptidyl transferase reaction did impair CAT-tailing.

Essential revisions:

All three reviewers agreed that the paper has considerable merit in uncovering an unexpected role for Rqc1 in ubiquitylation, suggesting that Rqc2-dependent CAT-tailing is required for efficient ubiquitylation, and in suggesting that CAT-tail synthesis should be achievable with only a 60S-peptidyl-tRNA complex, charged Ala and Thr tRNAs and Rqc2, without inclusion of the translation GTPases. However, certain results need to be bolstered by additional experiments or quantification of data from replicate determinations. Two of the reviewers requested additional experiments to prove that the mass-shifted material corresponds to CAT-tailed species. Quantification of the peptidyl-tRNA and CAT-tailed species from replicate experiments is needed to bolster the conclusions that the Rqc2-D98A mutant promotes mild peptidyl-tRNA hydrolysis in Figure 1; and to demonstrate that GMPPCP does not affect peptidyl-tRNA hydrolysis in Figure 2. Two of the reviewers also requested that you describe in some detail the modifications and critical variables in the preparation of that extract required to reconstitute RQC activities. Finally, reviewer #1 asks you to consider an additional experiment in which aniosomycin is used to inhibit CAT-tailing and determine whether ubiquitylation is also reduced in the presence of WT Rqc2. If so, this would overcome the complication of a possible pleiotropic effect of Rqc2-D98A on both ubiquitylation and CAT-tailing. If this last experiment cannot be carried out or interpreted unambiguously, it seems necessary to include in the Discussion the possibility that the Rqc2-D98A mutation impairs ubiquitylation as well as CAT-tailing because it alters the structure of Rqc2 in a way that disrupts its scaffolding role in supporting ubiquitylation by Ltn1, in addition to disrupting CAT-tailing.

The authors also are asked to make suitable revisions to the text to address all of the less critical points raised by the referees.

Reviewer #1:

This study reconstitutes in yeast extracts for the first time both the CAT-tailing and ubiquitylation activities of the RQC and exploits translation elongation inhibitors and mutants lacking different RQC components to probe the mechanisms and interdependencies between these two reactions. At odds with previously published work, they find that Rqc1 is required for ubiquitylation of the stalled peptides generated in their system using a non-STOP mRNA template, being defective in the rqc1D extract and rescued with purified Rqc1. As expected from the literature, CAT-tailing is defective in the rqc2Δ extract and can be rescued with purified WT Rqc2, but not with the D98A mutant. Unexpectedly, however, they find that the Rqc2-D98A substitution, in addition to deletion of the protein, impairs ubiquitylation as well as CAT-tailing, whereas previous studies had indicated no impact of this mutation on degradation of aberrant nascent chains in yeast cells, nor a requirement for Rqc2 in ubiquitylation of nascent chains in a mammalian extract. The authors propose instead that CAT-tailing and ubiquitylation are interdependent activities. They are also able to probe the effects of elongation inhibitors on the mechanism of CAT-tailing by the RQC complex using the preassembled, arrested 60S-peptidyl-tRNA complexes and find no effect of 40S inhibitors, as expected; nor of cycloheximide, which suggests no role for E-site binding of deacylated tRNA in CAT-tailing. Inhibitors of the translation GTPases also had no effect, eliminating requirements for EF1 and EF2 in recruitment of charged Ala or Thr tRNAs and translocation, respectively; whereas inhibitors of the peptidyl transferase reaction did impair CAT-tailing.

These findings are significant in uncovering an unexpected role for Rqc1 in ubiquitylation, suggesting that Rqc2-dependent CAT-tailing is required for efficient ubiquitylation, and in suggesting that CAT-tail synthesis could be achievable with only a 60S-peptidyl-tRNA complex, charged Ala and Thr tRNAs and Rqc2 without the translation GTPases.

Critical comments: My main concern is whether their finding that the Rqc2-D98A variant impairs ubiquitylation as well as CAT-tailing might arise because the D98A mutation is not as specific as believed and alters the 3-D structure of Rqc2 in a way that disrupts its scaffolding role in supporting ubiquitylation by Ltn1, in addition to disrupting CAT-tailing. If so, this would undermine the significant conclusion that CAT-tailing promotes ubiquitylation. It seems possible that they could conduct an incisive test of their model, that the efficiency of ubiquitylation is stimulated by CAT-tailing, by using aniosomycin to inhibit CAT-tailing (as they showed is possible) and then show that ubiquitylation is reduced in the presence of WT Rqc2, which would overcome the complication of a possible pleiotropic effect of Rqc2-D98A on both ubiquitylation and CAT-tailing.

Reviewer #2:

The work of Osuna et al., explores the quality control mechanism, RQC, that operates in yeast as well as other eukaryotes. The previous groundbreaking work from the Frost lab has shown that after splitting of the stalled ribosome, the nascent chain, associated in the form of peptidyl-tRNA with the large ribosomal subunit, is elongated in a non-templated manner with the "CAT" tails composed of Ala and Thr residues. The CAT-tail addition was shown to be promoted by Rqc2, which interplays with the activity of other proteins of the Rqc2 complex is releasing the nascent chain and targeting it for ubiquitination and degradation. Many fundamental questions about operation of RQC remain unanswered and the current work addresses some of those questions.

The development of an *S. cerevisiae* cell-free translation system, which allowed authors recapitulate both the CAT-tail formation and nascent chain ubiquitination is an important technical advancement. Using this system the authors have generated several important new insights by showing that: 1) the peptidyl transferase center is involved in the CAT-tail formation, but likely not the translation elongation factors; 2) Rqc1 is directly involved in the nascent chain ubiquitination; 3) CAT-tailing stimulates the ubiquitination of at least some nascent chains.

The study is well performed and well presented. Most of the conclusions are convincingly supported by the experimental data. Few points, however, need clarification.

Specifically:

1) The authors indicate that their experiments have been made possible by optimizing the in vitro translation system. However, they never commented what exactly has been optimized in comparison with the previously available protocols.

2) Subsection “A cell-free system that recapitulates Rqc2p-mediated nascent-chain elongation” and Figure 1 (δ rqc2 panel). The upper band is probably indeed peptidyl-tRNA, but this conclusion would be much more convincing if the authors have shown that the upper band disappears upon RNase treatment.

3) The authors present several fairly convincing, but rather indirect arguments that the 'mass-shifted' material is generated by CAT-tailing. Demonstrating either by Northern blotting the presence exclusively of tRNA(Ala) and/or tRNA(Thr) in the 'peptidyl-tRNA' band or by showing the incorporation of radioactive Ala or Thr residues in the tails would make this claim much stronger.

4) Subsection “A cell-free system that recapitulates Rqc2p-mediated nascent-chain elongation”andFigure 1. The relative intensity of peptidyl-tRNA and full product bands in the D98A mutant is comparable to that in the 'no Rqc2' control. Therefore, the claim that the Rqc2 mutant 'promotes mild peptidyl-tRNA hydrolysis' is questionable. Quantifying the bands (in several experimental replicates) and presenting the results of the quantification could make this claim stick.

Reviewer #3:

The authors report the development of a yeast (*Saccharomyces cerevisiae*) in vitro translation system that recapitulates activities of the ribosome quality control complex. Translating an HA-NanoLuc reporter lacking a stop codon and lysine residues, they demonstrate generation of species that behave like a nascent peptide with a covalent attachment to a peptidyl-tRNA and like the C-terminally tailed protein species. Conversion of the peptidyl-tRNA to free peptide and generation of the C-terminally tailed protein species require Rqc2, Dom34 and Hbs1 proteins, consistent with their in vivo functions. The authors demonstrate that addition of purified Rqc2 protein allows addition of the C terminal tail and peptidyl tRNA hydrolysis. They provide direct evidence that addition of the C terminal tail is mechanistically distinct from canonical translation; it depends upon the 60S subunit of the ribosome, but not upon the 40S. They recapitulate Ltn1-dependent ubiquitination. While most of these results are expected based on in vivo analysis and other results, they are crucial to establish the validity of the system for further analyses. Furthermore, the authors also provide evidence that Rqc1 plays a role in Ltn1-mediated ubiquitination and that there is an interplay between CAT tailing and ubiquitination.

The development and characterization of the yeast system is an important contribution to the field because it allows the coupling of genetics with the biochemical analysis. Moreover, the observations about the roles of Rqc1 and CAT tailing in promoting ubiquitination are important. However, there are some points that should be addressed by the authors.

Critique:

1) The authors consistently refer to the C-terminal additions to their polypeptide as CAT tails, but offer no evidence of this point. The Shen et al., 2015 paper used amino acid analysis as well as Edman degradation to demonstrate the addition of alanine and threonine to their in vivo reporter. This specificity is a key aspect of the RQC in yeast and differs from the mammalian system- it should be proven.

2) In the demonstration that C-terminal additions differ from canonical translation, the authors claim that GMP-PCP (the non-hydrolyzable GTP analog) had no effect on tailing. However, this does not appear consistent with the result in the Figure 2-in the lane with GMP-PCP, the peptidyl-tRNA band is nearly identical to the no Rqc2 lane, indicating a defect in peptidyl-tRNA hydrolysis. Also, how do the authors know that the GTP in the extract is depleted in this experiment?

3) The authors state that Rqc1 is "directly involved in nascent chain ubiquitination" (subsection “Ltn1p- and Rqc1p-dependent ubiquitination in the yeast cell-free system”). It might be worth determining if the role of Rqc1 is really essential or can be substituted by addition of a higher concentration of Ltn1 protein. While the Brandman et al., paper stated that "inclusion of Ltn1, Rqc1, and Tae2 in the RQC was not abolished by deletion of other RQC components", it is possible that the efficiency of recruitment of the RQC is dependent upon interactions. If this is the case, a higher concentration of Ltn1 might substitute for Rqc1.

4) One of the major contributions of this paper is the "optimized" in vitro translation system. It would add a lot to the paper to describe the modifications and critical variables in the preparation of that extract

---

## [Author Response]

*Essential revisions:*

*All three reviewers agreed that the paper has considerable merit in uncovering an unexpected role for Rqc1 in ubiquitylation, suggesting that Rqc2-dependent CAT-tailing is required for efficient ubiquitylation, and in suggesting that CAT-tail synthesis should be achievable with only a 60S-peptidyl-tRNA complex, charged Ala and Thr tRNAs and Rqc2, without inclusion of the translation GTPases. However, certain results need to be bolstered by additional experiments or quantification of data from replicate determinations.*

*Two of the reviewers requested additional experiments to prove that the mass-shifted material corresponds to CAT-tailed species.*

We share the reviewers’ sentiment and would have liked to demonstrate that the untemplated extensions we observe in vitroare indeed composed of alanine and threonine. We have tried to accomplish this with a variety of technically challenging assays but have not been successful (more below). Fortunately, none of our conclusions depend on knowing the precise composition of the Rqc2p-dependent extensions. In the revised text, we now explicitly acknowledge the uncertainty about their amino acid composition and note that we only refer to the untemplated Rqc2p-dependent C-terminal extensions as “CAT tails” for simplicity. Nevertheless, we suspect that the extensions are bona fide CAT tails based on their having the two other characterized properties:

1) The extensions require Rqc2p and the PTC of the 60*S* subunit but not Ltn1p, Rqc1p, or the 40*S* subunit (Figure 1, Figure 2).

2) The extensions are untemplated and located at the C-terminus (Figure 1 and Figure 3).

The alanine/threonine composition of the extensions is the only property of CAT tails that we were not able to demonstrate due to technical limitations of our system. Please refer to point 3) under “Reviewer #2” and point 1) under “Reviewer #3” for further explanation as to why the suggested experiments have not been successful or feasible. In addition to the experiments suggested by the reviewers, we have also tried the following other strategies:

Depleting Ala/Thr-charged tRNAs from CAT-tailing reactions by adding the amino alcohols of alanine and threonine (alaninol and threoninol), which inhibit tRNA charging (Hansen, Vaughan and Wang, JBC 1972; Arnez, Dock-Bregeon and Moras, JMB 1999). However, addition of any amino alcohol (e.g., serinol or isoleucinol) resulted in nonspecific inhibition of the in vitro translation reactions.

Depleting Ala/Thr-charged tRNAs from CAT-tailing reactions by omitting alanine and threonine from the ScIVT reactions. Surprisingly, omission of any one amino acid did not globally inhibit translation or CAT tailing. We suspect this is due to insufficient amino-acid depletion from the extract during dialysis, sufficient amounts of endogenous charged tRNAs in the extract, or perhaps misincorporation.

Generating more mass-shifted products for total amino acid analysis or Edman degradation. Despite our best efforts, we have not been able to generate extracts that produce enough material for these kinds of analyses (which we used successfully in the past, but only with Coomasie-stainable amounts of material obtained from in vivo yeast studies).

*Quantification of the peptidyl-tRNA and CAT-tailed species from replicate experiments is needed to bolster the conclusions that the Rqc2-D98A mutant promotes mild peptidyl-tRNA hydrolysis in Figure 1; and to demonstrate that GMPPCP does not affect peptidyl-tRNA hydrolysis in Figure 2.*

In the revised text, we have removed the conclusion that the Rqc2p-D98A mutant promotes mild peptidyl–tRNA hydrolysis. Given the recent revelation that eIF5A globally stimulates translation elongation and termination (Schuller et al., Mol Cell 2017), we cannot rule out the possibility that eIF5A, rather than Rqc2p-D98A alone, promotes the mild hydrolysis we observed (among other possibilities). With regards to GMP-PCP, we have removed any claims regarding the effect of GMP-PCP on peptidyl–tRNA hydrolysis. Though more peptidyl–tRNA accumulates in reactions supplemented with GMP-PCP in comparison to reactions supplemented with GTP, this may be due to inhibition of the GTPase Hbs1p, which facilitates ribosome splitting in concert with Dom34p. The persistent peptidyl–tRNA in the presence of GMP-PCP likely represents a population of 80*S* ribosomes that were not yet split at the time of GMP-PCP addition. Despite this, CAT tailed species still accumulate in the presence of GMP-PCP, indicating that GTP hydrolysis is not required for CAT tailing to occur on the available pool of 60*S*:peptidyl–tRNA.

Two of the reviewers also requested that you describe in some detail the modifications and critical variables in the preparation of that extract required to reconstitute RQC activities.

We have now included more details about the critical modifications for preparation of robustly active translation extracts that reconstitute RQC activities.

*Finally, reviewer #1 asks you to consider an additional experiment in which aniosomycin is used to inhibit CAT-tailing and determine whether ubiquitylation is also reduced in the presence of WT Rqc2. If so, this would overcome the complication of a possible pleiotropic effect of Rqc2-D98A on both ubiquitylation and CAT-tailing. If this last experiment cannot be carried out or interpreted unambiguously, it seems necessary to include in the Discussion the possibility that the Rqc2-D98A mutation impairs ubiquitylation as well as CAT-tailing because it alters the structure of Rqc2 in a way that disrupts its scaffolding role in supporting ubiquitylation by Ltn1, in addition to disrupting CAT-tailing.*

This is an excellent suggestion, and we have conducted the proposed experiment with anisomycin using two different sets of controls. The results of these experiments are presented in the new Figure 4 (using a lysine-free mRNA as a negative control) and Figure 4—figure supplement 3 (using the absence of Rqc2p as a negative control). We found that even in the presence of WT Rqc2p, ubiquitination dramatically decreases when CAT tailing is inhibited with anisomycin. This new data supports the conclusion that CAT tailing directly enhances Ltn1p-mediated ubiquitination and rule out the possibility that the observed reduction in ubiquitination in reactions conducted with Rqc2p(D98A) is due to pleiotropic effects imparted by the D98A mutation.

*The authors also are asked to make suitable revisions to the text to address all of the less critical points raised by the referees.*

*Reviewer #1:*

*My main concern is whether their finding that the Rqc2-D98A variant impairs ubiquitylation as well as CAT-tailing might arise because the D98A mutation is not as specific as believed and alters the 3-D structure of Rqc2 in a way that disrupts its scaffolding role in supporting ubiquitylation by Ltn1, in addition to disrupting CAT-tailing. If so, this would undermine the significant conclusion that CAT-tailing promotes ubiquitylation. It seems possible that they could conduct an incisive test of their model, that the efficiency of ubiquitylation is stimulated by CAT-tailing, by using aniosomycin to inhibit CAT-tailing (as they showed is possible) and then show that ubiquitylation is reduced in the presence of WT Rqc2, which would overcome the complication of a possible pleiotropic effect of Rqc2-D98A on both ubiquitylation and CAT-tailing.*

We have incorporated this excellent experimental suggestion into the revised manuscript in the new Figure 4 and Figure 4—figure supplement 3. Both experiments support the conclusion that CAT tailing directly enhances Ltn1p-mediated ubiquitination, as described in point #4 under “Essential Revisions”. Moreover, we think that the D98A mutation is unlikely to impair Rqc2p’s scaffolding role for Ltn1p because D98 is found in the NFACT-N domain, which our prior work established is responsible for binding the A-site tRNA (Shen et al., 2015). This domain is remote from the sarcin-ricin loop where Rqc2p meets Ltn1p on the 60*S* subunit. In addition, we showed in our prior work that a structure-based mutant of the NFACT-N domain, which contained the D98A substitution, still assembled normally with the 60*S* subunit and with Ltn1p but was defective for CAT-tailing activity in vivo.

*Reviewer #2:*

*1) The authors indicate that their experiments have been made possible by optimizing the* in vitro *translation system. However, they never commented what exactly has been optimized in comparison with the previously available protocols.*

We thank the reviewer for encouraging us to be more detailed about our protocol and have included a new emphasis on the critical modifications for preparing translation extracts that reconstitute RQC activities. Please refer to point 3) under “Essential Revisions” for further details.

*2) Subsection “A cell-free system that recapitulates Rqc2p-mediated nascent-chain elongation” and Figure 1 (δ rqc2 panel). The upper band is probably indeed peptidyl-tRNA, but this conclusion would be much more convincing if the authors have shown that the upper band disappears upon RNase treatment.*

Though we did not RNase-treat reactions conducted with *rqc2Δ* extracts to demonstrate that the ~43 kDa upper band is peptidyl–tRNA, we instead showed in Figure 2 that this species completely collapses upon treatment with puromycin. The puromycin sensitivity of the upper band unambiguously demonstrates that this species is, in fact, peptidyl–tRNA. This finding is consistent with the RNase sensitivity of the upper band (of similar molecular weight) observed at early time points in wild-type extracts, which is presented in Figure 1.

*3) The authors present several fairly convincing, but rather indirect arguments that the 'mass-shifted' material is generated by CAT-tailing. Demonstrating either by Northern blotting the presence exclusively of tRNA(Ala) and/or tRNA(Thr) in the 'peptidyl-tRNA' band or by showing the incorporation of radioactive Ala or Thr residues in the tails would make this claim much stronger.*

As noted above, we agree that determining the precise composition of the untemplated extensions would be valuable and we appreciate the reviewer’s experimental suggestions. Please refer to point 1) under “Essential Revisions” and point 1) under “Reviewer #3” for explanations as to why the request for this data present a formidable challenge.

We have attempted the suggested experiment to conduct ScIVT reactions in the presence of tritiated alanine; however, we have not been able to successfully detect radiolabeled translation products (either by exposing to a phosphorscreen or by scintillation). We surmise that this is likely due to the fact that our extracts do not generate enough protein product to surpass the limit of detection for ^3^H signal given that tritium has a very low specific activity (<50 Ci/mmol for [^3^H]-alanine). Further, even if we were able to demonstrate that radiolabeled alanine is incorporated into the untemplated extensions we refer to as CAT tails, this does not rule out the possibility that other amino acids are also incorporated into the extensions. Lastly, we were not able to find a source of radiolabeled threonine to use in ScIVT reactions to at least demonstrate that both alanine and threonine are present in the untemplated extensions.

We have also considered the suggested Northern blotting for tRNA(Ala) or tRNA(Thr) in the peptidyl–tRNA band. However, our finding that CAT tailing and peptidyl–tRNA hydrolysis are temporally coupled makes it difficult to stabilize CAT-tailing elongation intermediates linked to tRNA. We think the vast majority of the peptidyl–tRNA species we detect represent 60*S*:peptidyl–tRNA complexes that have yet to be engaged by the RQC. Thus, we would not expect that species to contain appreciable amounts of tRNA(Ala) or tRNA(Thr). We have tried many other experiments to demonstrate that the mass-shifted products are, in fact, composed of alanine and threonine, but none of these strategies have been successful due to the technical limitations of our system described above.

*4) Subsection “A cell-free system that recapitulates Rqc2p-mediated nascent-chain elongation”and Figure 1. The relative intensity of peptidyl-tRNA and full product bands in the D98A mutant is comparable to that in the 'no Rqc2' control. Therefore, the claim that the Rqc2 mutant 'promotes mild peptidyl-tRNA hydrolysis' is questionable. Quantifying the bands (in several experimental replicates) and presenting the results of the quantification could make this claim stick.*

We have removed the conclusion that the Rqc2-D98A mutant promotes mild peptidyl–tRNA hydrolysis. Please refer to point 2) under “Essential Revisions” for further details.

*Reviewer #3:*

*1) The authors consistently refer to the C-terminal additions to their polypeptide as CAT tails, but offer no evidence of this point. The Shen et al., 2015 paper used amino acid analysis as well as Edman degradation to demonstrate the addition of alanine and threonine to their* in vivo *reporter. This specificity is a key aspect of the RQC in yeast and differs from the mammalian system- it should be proven.*

Unfortunately, conducting amino acid analysis or Edman degradation requires a substantial amount of protein, beyond what our extracts are able to generate even after scaling up. While the specificity for alanine- and threonine-tRNAs is a key aspect of physiological CAT tails, none of our conclusions about the mechanism and function of CAT tailing are dependent on CAT-tail composition. Moreover, since CAT tails have yet to be observed in mammals, the composition of mammalian CAT tails remains unknown. Please see our earlier comments on this important point: point 1) under “Essential Revisions” and point 3) under “Reviewer #2”.

*2) In the demonstration that C-terminal additions differ from canonical translation, the authors claim that GMP-PCP (the non-hydrolyzable GTP analog) had no effect on tailing. However, this does not appear consistent with the result in the Figure 2-in the lane with GMP-PCP, the peptidyl-tRNA band is nearly identical to the no Rqc2 lane, indicating a defect in peptidyl-tRNA hydrolysis. Also, how do the authors know that the GTP in the extract is depleted in this experiment?*

We apologize for the confusion about our GMP-PCP results and did not intend to make any claims regarding the effect of GMP-PCP on peptidyl–tRNA hydrolysis. Though more peptidyl–tRNA accumulates in reactions supplemented with GMP-PCP in comparison to reactions supplemented with GTP, this is likely due to inhibition of the GTPase Hbs1p involved in ribosome splitting. Therefore, the persistent peptidyl–tRNA in the presence of GMP-PCP likely represents protection inside 80*S* ribosomes that were not yet split at the time of GMP-PCP addition. Nevertheless, CAT-tailed species still accumulate in the presence of GMP-PCP, suggesting that GTP hydrolysis is not absolutely required for CAT tailing to occur on the available pool of 60*S*:peptidyl–tRNA.

With regards to GTP depletion, most endogenous GTP is probably removed by the extensive dialysis that we perform during preparation of IVT extracts. Indeed, in pilot experiments we found that adding exogenous GTP was essential to detect any IVT products, suggesting that most endogenous GTP had been depleted. Though we add GTP back to IVT extracts (post-dialysis) to generate RQC substrate (i.e., 60*S*:peptidyl–tRNA), the concentration of GMP-PCP used in our experiments is sufficient to entirely inhibit translation when added at the beginning of the reaction (Figure 2, lane 3). Thus, we are confident that GMP-PCP is present at high enough concentrations to outcompete any remaining GTP and inhibit the translational GTPases.

*3) The authors state that Rqc1 is "directly involved in nascent chain ubiquitination" (subsection “Ltn1p- and Rqc1p-dependent ubiquitination in the yeast cell-free system”). It might be worth determining if the role of Rqc1 is really essential or can be substituted by addition of a higher concentration of Ltn1 protein. While the Brandman et al., paper stated that "inclusion of Ltn1, Rqc1, and Tae2 in the RQC was not abolished by deletion of other RQC components", it is possible that the efficiency of recruitment of the RQC is dependent upon interactions. If this is the case, a higher concentration of Ltn1 might substitute for Rqc1.*

We thank the reviewer for proposing this idea and have conducted the suggested experiment, which we present in the new Figure 4—figure supplement 2. We found that the role of Rqc1p in facilitating nascent-chain ubiquitination could not be substituted by a higher concentration of Ltn1p.

*4) One of the major contributions of this paper is the "optimized"* in vitro *translation system. It would add a lot to the paper to describe the modifications and critical variables in the preparation of that extract*

We thank the reviewer for encouraging us to be more detailed about our protocol and have included a new emphasis on the critical modifications for preparing translation extracts that reconstitute RQC activities. Please refer to point 3) under “Essential Revisions” for further details.